# Monsoonal imprint on late Quaternary landscapes of the Rub' al Khali Desert

Abdallah S. Zaki [1,2,3] ✉, Antoine Delaunay [4], Guillaume Baby[4], Negar Haghipour[5], Cécile Blanchet [6], Anne Dallmeyer [7], Pietro Sternai [8,9], Sam Woor[10,11], Omar Wani [2,12], Hany Khalil[13], Mathieu Schuster[14], Michael Petraglia [15,16,17], Florence Sylvestre[18], Giovan Peyrotty[1], Mohamed Ali[19], Frans Van Buchem[4], Abdulkader M. Afifi [4] & Sébastien Castelltort [1]

Abundant geomorphological, biological, and isotopic records show that Arabia repeatedly underwent significant climate-driven environmental changes during late Quaternary humid periods. Precisely mapping how the enhancement and expansion of the African Monsoon during these humid periods have affected landscape evolution and human occupation dynamics in Arabia remains a scientific challenge. Here we reconstruct an ancient water-sculpted landscape consisting of lake and river deposits, coupled with a large outlet valley in the Rub' al Khali Desert of Saudi Arabia. During the peak of the Holocene Humid Period or before, intense rainfall reactivated alluvial floodplains and filled a ~1100 km² topographic depression, which eventually breached, carving a deep ~150 km-long valley. Coupling geologic reconstructions with transient Earth system model simulations shows that this hydrological activity was linked to higher seasonal precipitation punctuated by repeated heavy events. Analysis of lacustrine and fluvial sedimentary deposits implies sediment routing across distances of up to 1000 km from the Asir Mountains. Our results indicate that such intense flooding challenges the conventional view of simple, weak, and linear landscape stabilization following increased rainfall in Arabia. Our findings highlight the crucial role of an enhanced African Monsoon in driving rapid landscape transformations in the Arabian Desert.

## Main

The Arabian Peninsula is mostly a vast arid to hyperarid desert region. However, abundant geological archives show evidence for an intermittently more humid past[1–4] (Fig. 1). During the early Holocene to middle Holocene (~11 to ~5.5 ka BP), perennial lakes and extensive drainage networks characterised Arabian landscapes[5–8]. Increased rainfall during the early Holocene, driven by the northward expansion of African and Indian Ocean monsoon rain belts[9–11], is the most recent of the multiple wetter periods that have occurred synchronously with insolation changes associated with orbital precessional cycles throughout Late Pleistocene to Holocene. Although this wetter period spanned the early to middle Holocene, its duration varied locally, extending over multiple millennia in the southern part of the peninsula[1–7] to just a few centuries in northern Arabia[8] due to the contribution of monsoonal systems. These wetter climatic conditions would have created grasslands and savannah-like environments, paving the way for significant human expansions across Arabia[11–13] (Fig. 1).

Multiple water isotope fingerprinting studies have found that the HHP intensification resulted from a reinforcement/strengthening of the Indian Monsoon, African Monsoons, and possibly Mediterranean Westerlies, given its situation at the confluence of multiple atmospheric circulations[3,10,14]. Some isotopic and model studies have suggested that the HHP in northern and western Arabia was associated with an eastward expansion of the African Humid Period[6,15,16], though there is questioning of the degree to which it resulted in increased precipitation[6]. There is still a fundamental debate about the extent and magnitude of the expansion of the African Monsoons into Arabia, which needs to be addressed to understand the relationships between climatic and environmental changes and the dynamics of human occupation patterns throughout the Holocene. Furthermore, understanding the influence of African monsoons on Arabian Desert landscapes is crucial for establishing benchmarks needed to accurately simulate the extent and magnitude of the Saharo-Arabian belt's greening[16–19].

The Rub' al Khali Desert is a large sedimentary basin, located downstream of the Asir Mountains (Fig. 1), characterized by aeolian dunes, drainage systems, and paleo lacustrine and palustrine deposits[5,12]. The Asir Mountains are hypothesized to have been primarily fed by the African

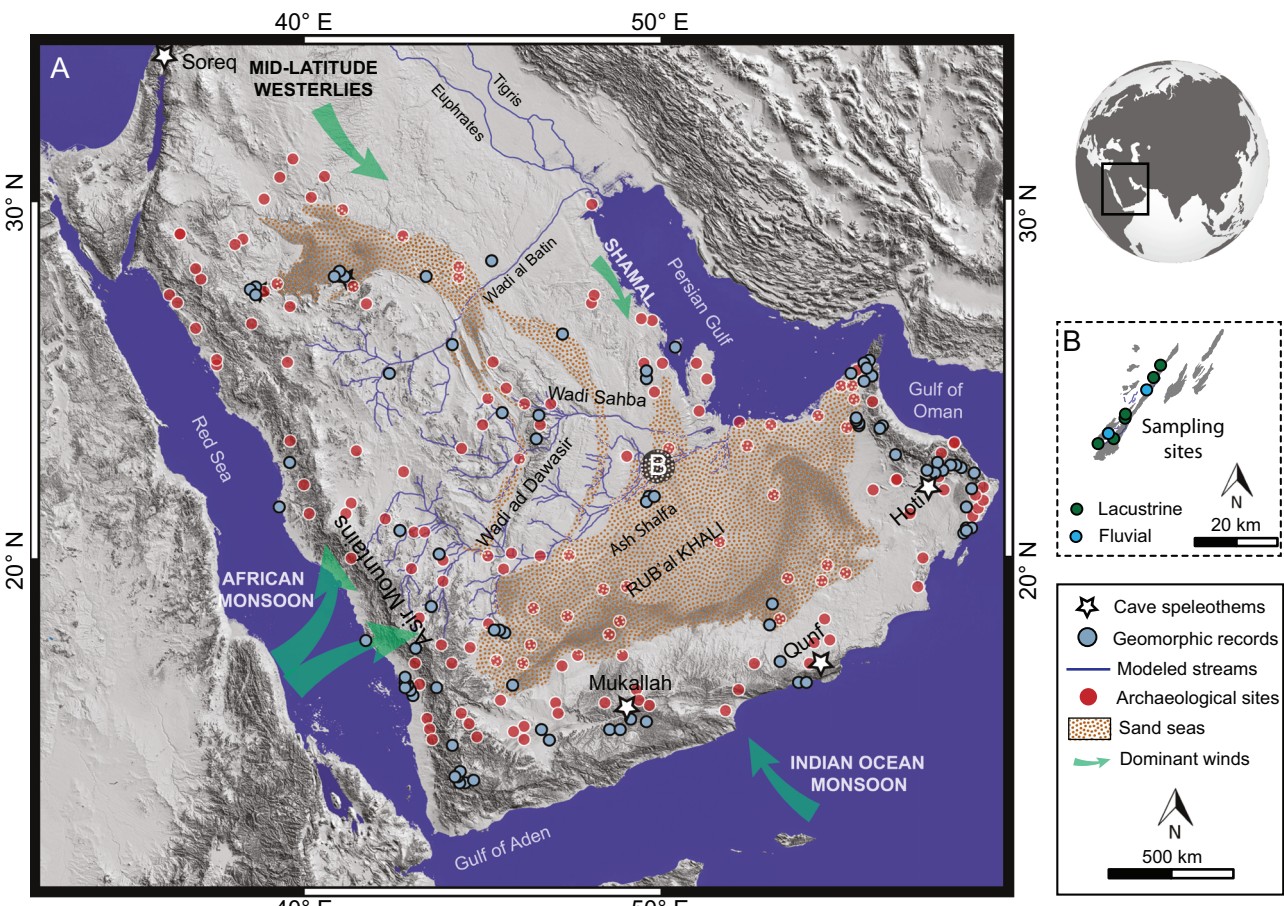

**Fig. 1 | Distribution of palaeohydrological and geomorphic records, archaeological sites, modeled streams, major monsoon systems, and the study site in Arabia. A** Hillshade map of Arabia highlighting the locations of the HHP palaeohydrological records (Supplementary data 1 and 2) and major archaeological sites[11]. The map displays the modeled streams, namely Wadi ad Dawasir, Wadi al Batin, and Wadi Sahba[5,12], as well as the dominant moisture-bringing atmospheric systems (monsoons, Westerlies). **B** A sketch map depicts the distribution of both the fluvial and the lacustrine deposits used in this study. The data are derived from a 30-m-resolution Shuttle Radar Topography Mission (SRTM) digital elevation model.

Monsoons[5]. Consequently, the drainage systems that originated and eroded upstream and deposited sediments downstream could serve as a proxy to constrain the magnitude and extent of the African Monsoon's influence on the Arabian Desert landscapes.

Here, we reconstruct an ancient water-sculpted landscape consisting of lacustrine, palustrine and fluvial deposits, coupled with a large outlet valley in the Rub' al Khali Desert of Saudi Arabia (Fig. 1). We used a multi-proxy approach incorporating quantitative remote sensing, field sedimentology, radiocarbon dating, isotopic provenance tracers, palaeohydraulic reconstructions, and lithospheric flexural and climate models. Our results illustrate the functioning of a large-scale source-to-sink fluvial system impacted by intense monsoons during the HHP, filling and breaching of a downstream topographic depression by water, and incision of a large outlet canyon. Furthermore, our results are useful for testing and improving Earth system models of the greening of the vast Arabian desert belt, which may help shed light on the environmental drivers of human population dynamics in the region.

## Geologic evidence of water-formed landscapes in the Rub'al Khali Desert

Due to its location downstream of large ancient drainage systems such as Wadi ad-Dawasir and Wadi Sahba (Fig. 1 and S1, S2), the Rub' al Khali Desert is a sensitive environment that records past hydroclimatic shifts in Arabia across various depositional environments, including fluvial, lacustrine and palustrine systems. Optical satellite images, combined with aerial images from an unmanned aerial vehicle (UAV) in the northeastern Rub' al

Khali near Ash Shalfa, reveal a distinct array of complex linear dunes (Fig. S3). These dunes are found beneath and between small, eroded patches that appear to be either wetland or floodplain lakes (Fig. S3), incised by channels of fluvial origin (Fig. S3). We observe both single-thread channel fills and braided channel belts (Figs. S2 and S3). The single-thread channels are mostly shallow, with widths not exceeding a few tens of meters, while the braided rivers preserve kilometer-scale channel belts (Figs. S2 and S3). These channels are incised within vast whitish mesas that we interpret to be remnants of small lakes or wetlands. Therefore, we interpret the region as preserving three different sedimentary depositional environments: aeolian, based on the complex linear dunes; fluvial, based on the incised and inverted channel fills and belts; and either lacustrine or palustrine, based on the eroded whitish mesas. Here, we focus on both the fluvial and lacustrine or wetland environments.

The deposits from either palaeolakes or palaeowetlands are topographically manifested as low mesa remnants, ranging from 0.15 m to 1.85 m in height across the landscape. These deposits exhibit stratigraphic similarities at five different locations, primarily consisting of massive fine sands to finely laminated, carbonate-rich marls (Fig. 2, Sections 1, 2, 4, 5, and 8). At one outcrop (outcrop 8), these deposits overlie laterally extensive loose sediments, which we interpret as aeolian sand (Fig. S4C). Additionally, at one site, we observe a 50-cm thick, undisturbed veneer of finely laminated marl deposits composed of fine silt and clay-sized particles (Fig. S4A and B). We interpret the presence of carbonate-rich marl facies as evidence of both a lacustrine environment and wetland setting, which is consistent with published faunal analyses from the distal reach of Wadi ad-Dawasir[5].

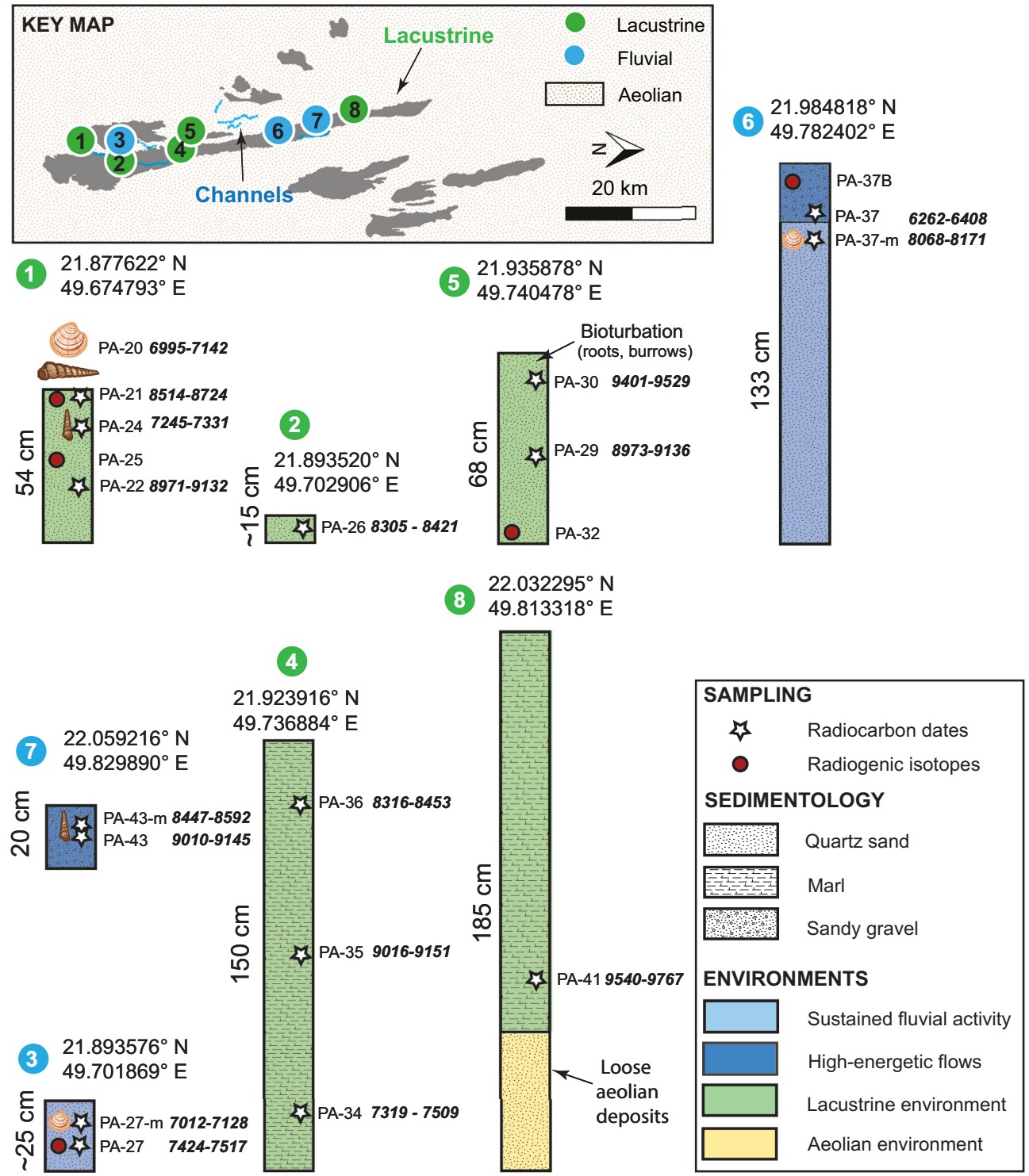

**Fig. 2 | Simplified stratigraphic columns of the eight examined sections in Ash Shalfa site, depicted alongside their locations on a geomorphic map derived from Landsat imagery and refined using aerial images.** Five sections contain lacustrine deposits and three sections consist of fluvial deposits. Locations of samples taken for radiocarbon and radiogenic isotope analysis are noted. The ages were calibrated using the IntCal20 curve (see Methods and SI for details) (Figs. S8 and S9).

Furthermore, three of the five outcrops display trace fossils, primarily burrows, indicating shallow-water zones or small lakes and ponds (Fig. S5)[20]. These findings align with the sedimentological characteristics of Pleistocene and Holocene lakes, where lacustrine deposits overlie aeolian sands, as seen in the Rub' al Khali, Jubbah, Tayma, and Mundafan in Saudi Arabia, Wahalah and Awafi in the UAE, and Wahiba Sands in Oman[5,21–23], suggesting recurrent similar climatic and depositional conditions throughout Arabia.

The palaeoriver deposits are discernible in the landscape as abandoned braided channels, sinuous channels, or sinuous inverted channels. We examined three outcrops, with thicknesses ranging from 0.2 m to 1.33 m (Sections 3, 6, and 7 in Fig. 2). We observed two distinct depositional patterns. The first consists of fine sands lacking any sedimentary structures, as well expressed in Sections 3 and 6 (Fig. 2). In these sections, and also in Section 7 (Fig. 2), there are thin layers ( ~ 0.2 m) of pebbles mixed with medium to coarse-grained sand. This variation suggests two different

depositional environments, ranging from sustained fluvial activity for the fine materials to high-energy flows indicated by the presence of sandy gravels.

We find that both lacustrine and fluvial deposits feature two types of well-preserved freshwater mollusks: *Unio tigridis* and *Melanoides tuberculatus* (Fig. S6)[24], indicating the presence of depauperate floodplain lakes in the northeastern region of the Rub' al Khali. Some of the *Unio tigridis* were observed with both valves intact, which possibly indicates that they were not transported, or at least were only transported very short distances, after death. This is consistent with observations at sites downstream of Wadi ad-Dawasir, documented by Matter et al.[5] where they observed specimens of *Unio tigridis*, indicating in-situ death in a freshwater environment.

Collectively, observations from large-scale satellite and aerial imagery, coupled with outcrop-scale observations, suggest a palaeolandscape composed of floodplains interspersed with waterbodies including lakes, ponds and wetlands a few tens of square kilometers in area. This interpretation is further supported by sedimentary thin-section petrography, which shows abundant carbonate clasts reworked in a fluvial channel (Fig. S7). This indicates that some of the fluvial deposits were reworked from extensive lacustrine areas. Such changes could be consistent with the varying extents of both fluvial and lacustrine environments over time in response to precipitation patterns. These findings further suggest that the region experienced stable, freshwater conditions that supported a low-energy aquatic ecosystem, consistent with the presence of *Unio tigridis* and *Melanoides tuberculatus*. The intact condition of *Unio tigridis* specimens implies minimal post-mortem transport, reinforcing the interpretation of in-situ freshwater environments. The co-occurrence of these species points to a dynamic palaeolandscape where floodplain lakes, wetlands, and fluvial systems fluctuated with shifting precipitation patterns, providing critical habitats during humid periods.

## Geomorphologic evidence of lake development and breach-induced flooding

The drainage networks and inverted channels that overlap and intersect the lacustrine deposits converge towards a depression located near Umm Athelah, 90 km east of the sampling sites (22.997070°N, 49.271578°E). Based on the geologic mapping, the depression geologically formed within Miocene and Pliocene sandstone, marl, and limestone, which are overlain by Quaternary gravels. The depression currently preserves Quaternary sabkha deposits at its base, primarily composed of saline silt. The exact origin of the depression remains uncertain. However, given the presence of faults and evidence of tectonic uplift in the surrounding region (Fig. S10)[25], it is plausible that tectonism played a role in its formation. This could have been followed by other surface processes such as aeolian erosion, karstification, and periodic fluvial activity, which may have contributed to shaping the depression over time. We interpret the depression to host large water body forming a lake as it is bordered by a distributary fan and connected to a valley ~150 km in length (Fig. 3). The valley exhibits geomorphic features such as streamlined islands, amphitheater-shaped heads, and a linear main channel (Figs. 3 and S11), which suggest deep and energetic flows associated with erosion and sediment transport. The upstream projection of the valley slope connects the valley to the topographic rim of the depression. Taken together, these observations suggest that valley formation is associated with the breach of the lake margin and subsequent overflow (Fig. S12). These observations and interpretations align morphologically and morphometrically with widely documented examples from flume experiments (Fig. S13)[26], and natural systems across the globe[27–30].

Our observations of the features within the depression reveal diverse features that we interpret as shoreline morphologies (Fig. S14). These morphologies include 70-m-high cliffs, a ~25-m-thick delta that preserves a pronounced slope break from the delta topset to the lower delta zone, and terraces (Fig. 4). Though no direct field observations from this site have been made, the presence of these features together provides additional evidence supporting the interpretation that a body of water once occupied this depression. This interpretation has been cross-validated through a quantitative classification and detection of landforms (see Methods). However, there is a possibility that these cliffs were originally formed as a result of regional tectonic uplift, and subsequently modified by water and wind activities. We have identified rhythmic patterns in the elevations of these morphologies, especially in the delta topset. This delta topset maintains an elevation range similar to one of the terraces, with elevations ranging from approximately 128 to 141 m asl (Fig. 4). Taken together, our observations and quantitative analyses robustly support the hypothesis that a paleolake was once present within this depression.

We calculated the volume of water required to fill the lake to the level of the lower zone delta (128 m above sea level (asl)), delta topset (141 m asl), and the lower incision level at the breach (148 m asl; Figs. S12 and S15). We computed the volumes of the lake at three different levels using 30-m-resolution SRTM data. Our findings revealed that at an elevation of 128 m asl, the lake would have contained $4.7 \times 10^9$ m³ of water within an area of 439 km². To reach a level of 141 m asl (the delta topset), it would have contained $1.38 \times 10^{10}$ m³ of water. Ultimately, the volume of water would have increased to $2.07 \times 10^{10}$ m³ at an elevation of 148 m asl (the point at which the outlet valley began to form) (Fig. 2). As the potential energy released during the breach is proportional to the total water volume drained[29], we quantified the energy associated with breach flooding (see Methods). Our findings indicate that the released potential energy was ca $3.14 \times 10^{15}$ J, which likely eroded $1.98 \times 10^{10}$ m³ of sediment, carving the outlet valley (Fig. 3). When comparing this event with other terrestrial events[29], including flume experiments and natural systems, this event appears to be remarkable among the most energetic occurrences documented in geologic history (Fig. 5).

One of the key questions is whether a single flood event was responsible for forming this outlet valley. Terraces within outlet valleys are important in determining whether their formation resulted from a singular event or multiple large episodes of flood events[26,27,30]. Each flooding event, characterized by its unique discharge volume, would result in terraces of varying depths, indicating multiple flooding episodes. Using an automated terrain classification algorithm, we have identified at least one terrace system within the valley (Fig. S16). This finding supports the hypothesis that the formation of the outlet valley occurred through episodic events. Such episodic, intense floods are prevalent in current-day deserts and would have been even more frequent during the HHP.

## Time-space evolution of lake systems

While previous research has documented the existence of Holocene lakes both on the periphery and within the Rub' al Khali Desert, including those in Mundafan, the Wadi ad Dwasir floodplain lakes, and the Awafi and Wahalah lakes[5,31–33], our study adds a new perspective in the investigation of the role of African Monsoon into HHP time and space evaluation. We have identified a previously unknown lake that is significantly larger—approximately 20 times the size of the Holocene Mundafan lake—associated with major breach flooding events. These events formed an outlet valley approximately 150 km long, likely the result of multiple events of heavy rainfall. Our research combines in-situ analysis from the Wadi ad Dwasir floodplain lakes and river deposits (serving as the drainage system for this large lake) and remote sensing observations and analyses of both the large lake and its outlet valley. Through these analyses of both the drainage system and the lake, we have identified two distinct phases in the Holocene evolutionary history of the Rub' al Khali lakes (Fig. 6).

The three lake levels we identified were deduced from geomorphic evidence, including delta topset, lower delta zone, and the breach point of the outlet valley. This evidence enabled us to extrapolate three possible independent water elevations at 128, 141, and 148 m asl, respectively (Figs. 3 and 4). We presented multiple lines of evidence supporting the presence of a lake, such as terraces and cliffs that preserve shorelines, and an outlet valley (Figs. 3 and S11-S16), though no field work was conducted here. Field observations and samples were, however, collected from floodplain deposits and lakes situated 90 km southeast of the main lake (Fig. 6). Across the intervening distance, we mapped large swathes containing

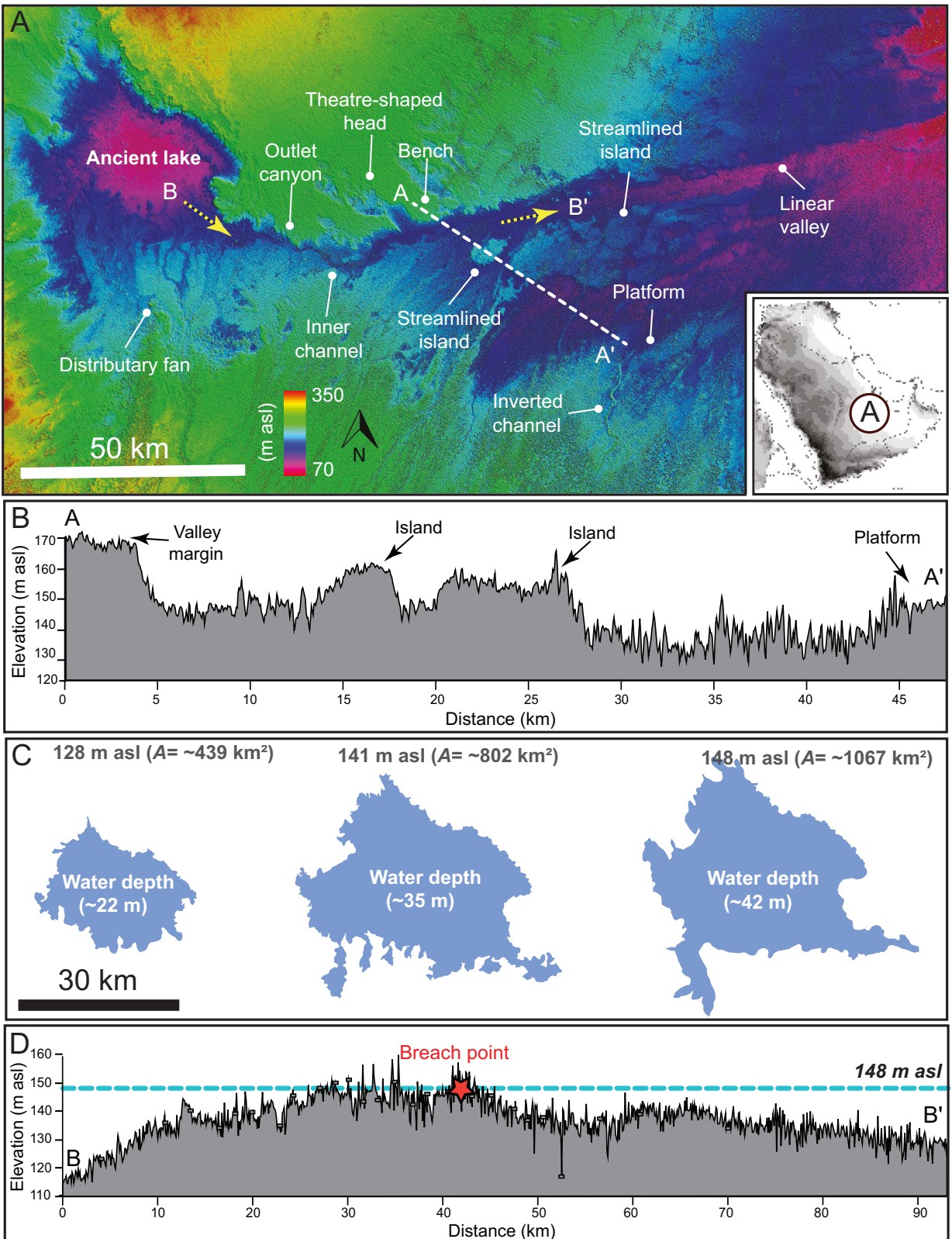

**Fig. 3 | Geomorphic indicators of breach-driven floods. A** Digital elevation model illustrating a depression, which we interpret to be an ancient lake, bounded by a distributary fan and connected via an outlet valley. Theatre-headed channels and streamlined islands within the valley indicate lake breach flooding. **B**, **D** Cross-sectional and longitudinal thalweg profiles derived from SRTM data. **C** Water volume modelling within the lake, pre-and post-flood. For reference, at 148 m asl, the lake was about 40 km in diameter.

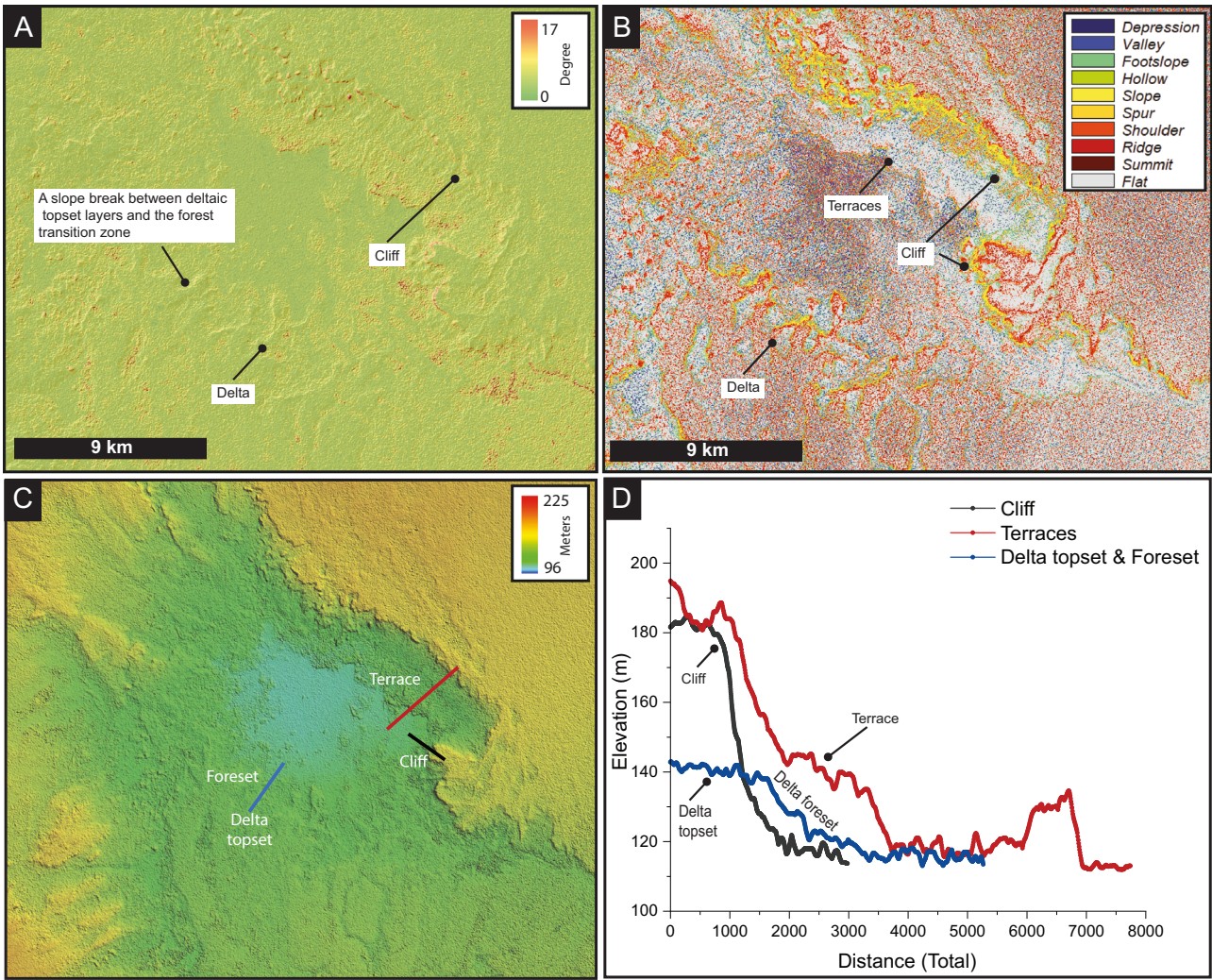

**Fig. 4 | Slope and geomorphic classification maps with cross-sectional profiles showing the delta, cliffs, and terraces of the proposed lake. A** Slope map of the depression that hosted the lake, indicating a delta dipping into the depression. This map also reveals that the northern flank of the depression is characterized by cliffs. **B** Geomorphometric map generated by the Geomorphons algorithm, which identifies both the high slopes associated with the cliffs and the elevated areas of the delta and adjacent forest, as well as the flat surfaces likely representing preserved terraces. Panels **C** and **D** display cross-sectional profiles of the observed landforms, including the delta, cliffs, and terraces. The terraces and delta feature consistent elevations, which could possibly indicate that both landforms were active at same time.

various patches of white color, which were systematically associated with probable lacustrine deposits in the Shalfa area. These deposits are trapped beneath and between modern dunes and extend over tens of kilometres. The patches were mapped using Landsat 8 and 9 multispectral imagery based on their colour and texture (Fig. 6C). Our criterion is the presence of a whitish deposit with a smooth texture in these images, similar to those identified at the field site (Figs. 6C and S3B, D). The presence of these deposits, along with stream channels (Figs. 6C and S2, S3), indicates that large floodplain lakes once occupied this region and were clearly connected to the main lake and its outlet valley. The suggestion of floodplain lakes is consistent with previous morphological, sedimentological, and palaeoecological reconstructions from the same and nearby regions[5]. Thus, we interpret these floodplain lakes and streams to represent the catchment of the main lake and its outlet valley system.

The radiocarbon ages of the 17 samples, consisting of organic materials (Supplementary data 3; Fig. 2), range from 9540–9767 cal. yr BP to 6262–6408 cal. yr BP. The oldest sample is from a lacustrine deposit 1.27 metres beneath the surface and dated at 9540–9767 cal. yr BP. The ages of the subsequent lacustrine samples ranged from 9401–9529 cal. yr BP to 7245–7331 cal. yr BP. Fluvial deposits are typically younger than lake deposits, ranging from 9010–9145 cal. yr BP for an inverted channel atop a

lake deposit to 6262–6408 cal. yr BP for incised drainage networks. Although two lacustrine sections (Sections 4 and 5 in Fig. 2) show older ages superimposed on younger ones, this discrepancy could be due to carbon contamination, which may have led to the reworking of older sediments into younger deposits. However, our overall ages are consistent with those reported locally or regionally (Supplementary data 1 and 2), indicating the presence of fluvial and lacustrine environments across the Arabian Desert. These fluvial and lacustrine ages align with existing data from the surrounding areas in Wadi Ad-Dawasir, obtained through radiocarbon and luminescence dating. Overall, the age sequence indicates a long-term shift in humidity, from lacustrine environments between 9540–9767 cal. yr BP and 7245–7331 cal. yr BP, to fluvial erosion, ending at 6262–6408 cal. yr BP and inversion under an increasingly arid climate by ca. 6000 cal. yr BP. This finding aligns with the extensive compilation of nearly 400 optically stimulated luminescence (OSL) and radiocarbon dates across the Arabian region. The statistical analysis of the lacustrine and fluvial populations of ages suggests that fluvial depositional environments are younger (ca. 5850 ± 480 cal. yr BP) than lacustrine environments (ca. 8300 ± 201 cal. yr BP) (Fig. S17).

The integrated evidence—including dating from the drainage basin, remotely sensed data from nearby inverted channels, and the geomorphic

**Fig. 5 | Comparison of flood volumes and depths for lake margin breach-related floods documented around the globe and in laboratory experiments[29], with the Rub' al Khali palaeolake breach.** The plot highlights the relationship between breached depth and drained volume, which are the primary determinants of the energy released by lake breach floods. The Rub' al Khali lake breach event is highlighted as one of the most energetic events compared to other terrestrial examples[27].

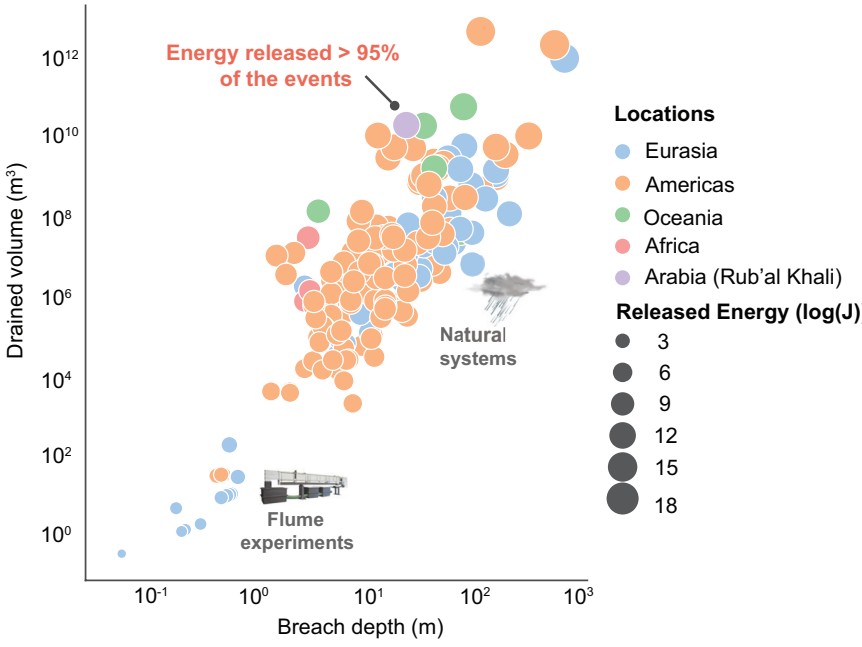

relationships of the outlet valley—suggests that the lake breach or high-stand events potentially occurred, or were reactivated, during the early Holocene (9540–9767 cal. yr BP to 6995–7142 cal. yr BP). However, we acknowledge the possibility that these features could have formed during Pleistocene humid periods, when wetter conditions prevailed in the Saharo-Arabian Desert belt.

The absence of well-preserved shorelines for these small floodplain lakes is likely due to substantial wind erosion, estimated at 20 ± 16 m from the top of the inverted channels in the region (Fig. S18). Additionally, thermomechanical geodynamic modeling, constrained by regional geophysical data (see Methods), suggests that flexural uplift of the surface elevation due to the abrupt draining of the lake, possibly up to 1.5 meters, may have further contributed to the erosion of ancient shorelines (Fig. S19).

Given these considerations, our findings suggest that the formation or reactivation of these hydrologic features could have occurred either during the early Holocene or earlier, possibly in the Pleistocene. Future studies focused on obtaining direct dating of these deposits will be essential to conclusively determine the timing and sequence of these geomorphic events.

## HHP rainfall patterns in the Rub' al Khali

Our study sites, both the floodplain and the main ancient lake, appear to have been primarily fed by Wadi ad Dawasir (Fig. 1). To further test this assumption, given most of the inferred streams originating from Asir mountains are modelled from digital elevation models (Fig. 1)[5,12], we used radiogenic neodymium (Nd) and strontium (Sr) isotopes to determine the provenance of five samples collected from the fluvial and lake sediments. Several samples contain a significant amount of a more radiogenic component, which tracks the input of volcanic material likely originating from the Asir Mountains near the Red Sea (Figs. 1 and S20; Supplementary data 4). Some samples contain a large amount of crystalline material, similar to the present-day aeolian dust from the region (Fig. S20), which might represent the remobilization of more local, surface sediments. Both geomorphic and geochemical evidence suggest that the lake was fed by precipitation occurring over the southern Red Sea region and draining into the study area (Fig. 1).

We explored the precipitation regime responsible for lake filling and breaching using Earth System model (ESM) simulations and palaeohydraulic constraints (see Methods)[34–36]. The ESM simulations identify the early Holocene as the wettest period in the region over the last 21,000 years

(Fig. 7; Movie S1). According to the MPI-ESM simulations, rainfall at the palaeolake location peaks at 8.7 ka with a positive precipitation anomaly of ~220 mm/year compared to the modern climate. The Trace21 ka-simulations indicate an even higher mean annual precipitation anomaly (~+300 mm/year compared to modern climate) during the HHP peak (at 9.1ka in these simulations). These estimates are in agreement with the geomorphic, sedimentologic, and geochronologic evidence from the extent and dynamics of lakes and floodplains in the Wadi ad Dawasir and Lake Mundafan, indicating a decrease in precipitation from approximately 9600 to 7900 cal.yr BP to around 7.9 to 4 ka BP (Fig. 8)[5]. In both models, the obtained positive rainfall anomaly results from the expansion of the West African monsoon rainbelt into the Sahel/Sahara region and large parts of Arabia. In the Trace21ka-ensemble, the annual mean precipitation surplus during the HHP even exceeds the increase in rainfall in North Africa. After the HHP peak, simulations show that the region experienced a strong decline in precipitation. At 6 ka, the annual rainfall amount had already decreased by 45% in the MPI-ESM ensemble ( ~ 120 mm/yr) and 35% in the Trace21ka-ensemble ( ~ 190 mm/yr) (Fig. 7).

Channel width and drainage area measurements were used to calculate channel-forming rainfall intensity (see Methods; Figs. 6C and S21). The calculated hourly rainfall intensities range from 17 ± 6 to 78 ± 27 mm/hr, and while we lack constraints to assess the duration and frequency of such events, it is remarkable that these rates are comparable to those that episodically punctuated eastern Sahara during the AHP[37,38]. Such intensities indicate that the delivery periods potentially persisted between ~4 to ~20 hr every year.

Integrating data from models and palaeohydraulics at different annual and hourly time scales suggests that the lake and its catchment area may have experienced sustained humid conditions. The mean annual precipitation was then ranging from approximately 220 to 300 mm/yr. Intense hourly precipitation (17–78 mm/hr) may have occurred intermittently during the HHP peak, with runoff pulses from the upstream catchment areas contributing to downstream lake breach flooding events.

Finally, the minimum duration required to reach the lake margin and begin to open the outlet valley can be estimated by dividing the minimum drained volume by the mean annual precipitation (MAP). This simple calculation suggests that the formation of the outlet valley would have taken between 67 and 91 years. However, this duration may be subject to an order-of-magnitude increase, as the drained volume considered here represents the minimum value. Additional uncertainties might arise from sediment

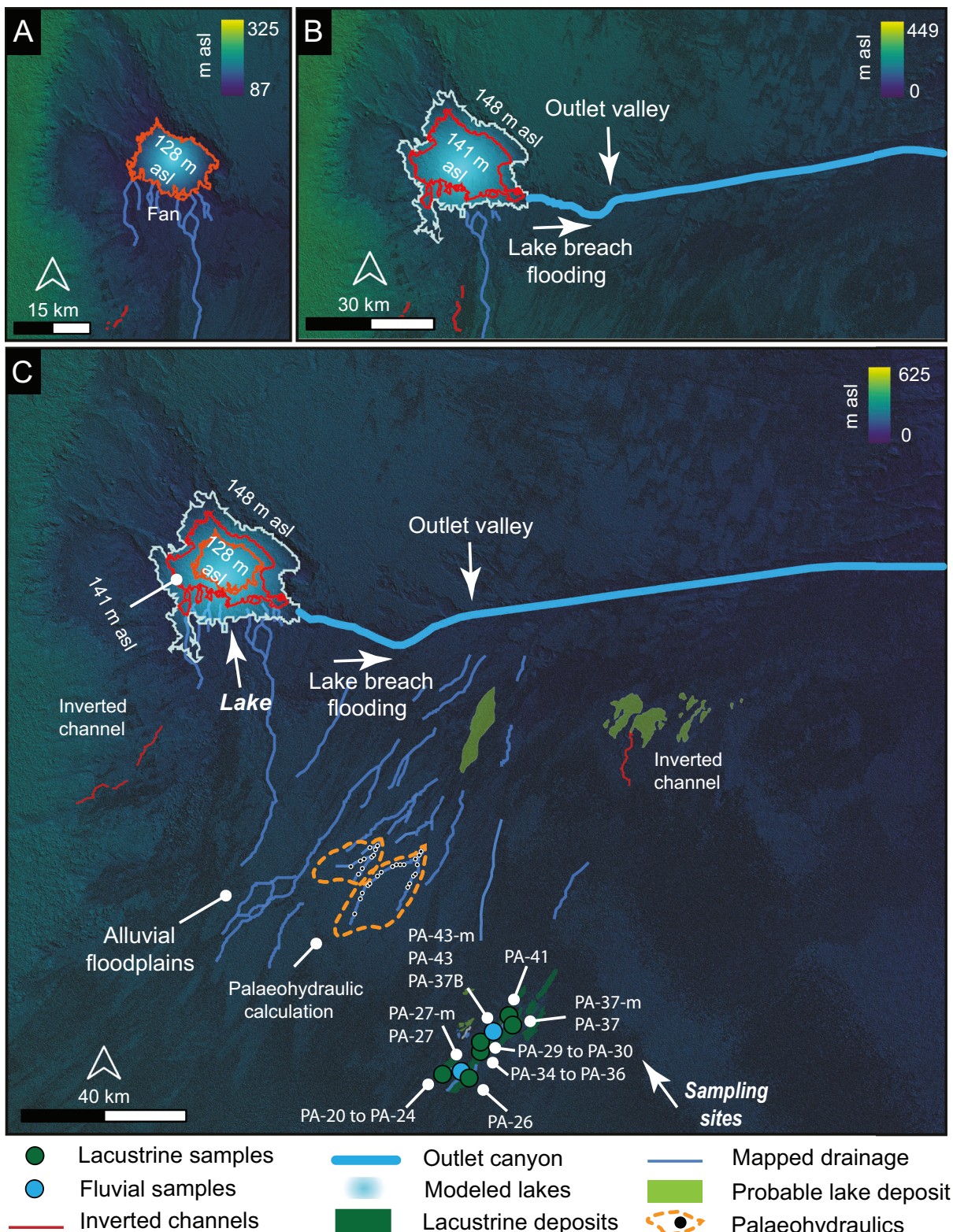

**Fig. 6 | Inferred palaeolake levels based on geomorphic observations. A** Delta fringing a depression at an elevation of 128 m asl for the lower delta zone and 141 m asl for the delta topset. **B** Coupling these observations with the existence of an outlet valley breaching at an elevation of 148 m asl supports a scenario of lake formation and subsequent breach-induced flooding. **C** Extensive lacustrine, probable lacustrine deposits, and fluvial deposits lack a discernible shoreline, suggesting the presence of floodplain lakes at an elevation of 202 m asl, which extend downslope towards the main lake and its outlet valley. The distance between the sampling site and the lake is approximately 90 km. Probable lacustrine deposits were identified using Landsat 8 and 9 multispectral imagery, based on their colour and texture. The background data are derived from a 30-metre-resolution SRTM digital elevation model.

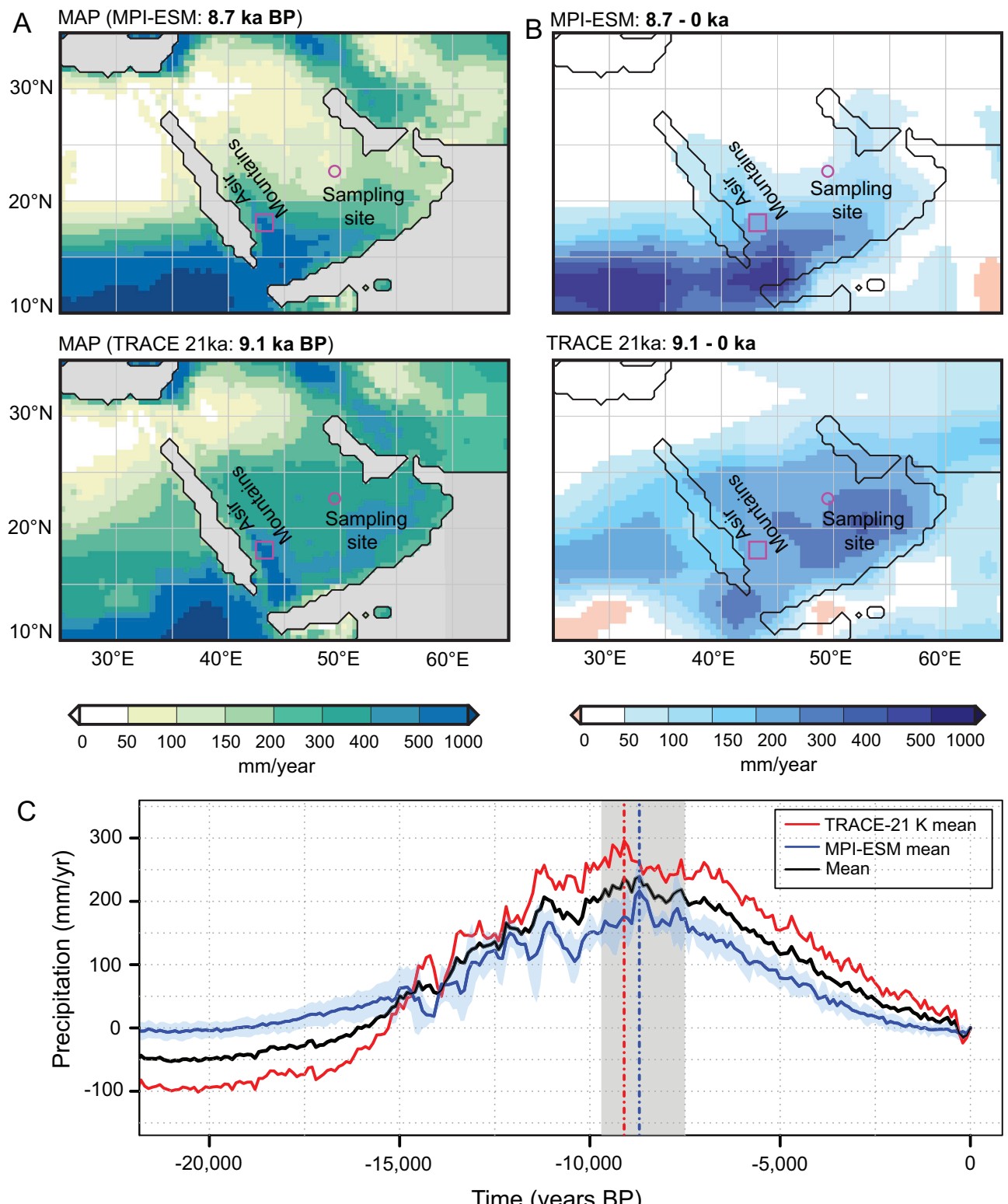

**Fig. 7 | Simulated mean annual precipitation at the peak of the Holocene Humid Period and precipitation patterns over the past 21,000 years. A** Simulated MAP absolute values at the simulated HHP peak: 9.1 ka BP (in TRACE-21K simulations) and 8.7 ka BP (in MPI-ESM simulations). **B** The difference in MAP values between the two time-slices, specifically from 9.1 ka BP (TRACE-21K) or 8.7 ka BP (MPI-ESM) to the present (0ka BP). **C** Illustrates the fluctuations in MAP over the past 21,000 years as an anomaly to 0ka BP in 100-year climatological means, highlighting that the early Holocene period experienced the most significant increase in precipitation. This finding is in line with our high-stand and breach flood records. The purple box in (**A**) and (**B**) indicates the Asir Mountains, while the circle denotes the sampling site. The dashed lines in **C** mark the timing of the HPP peak.

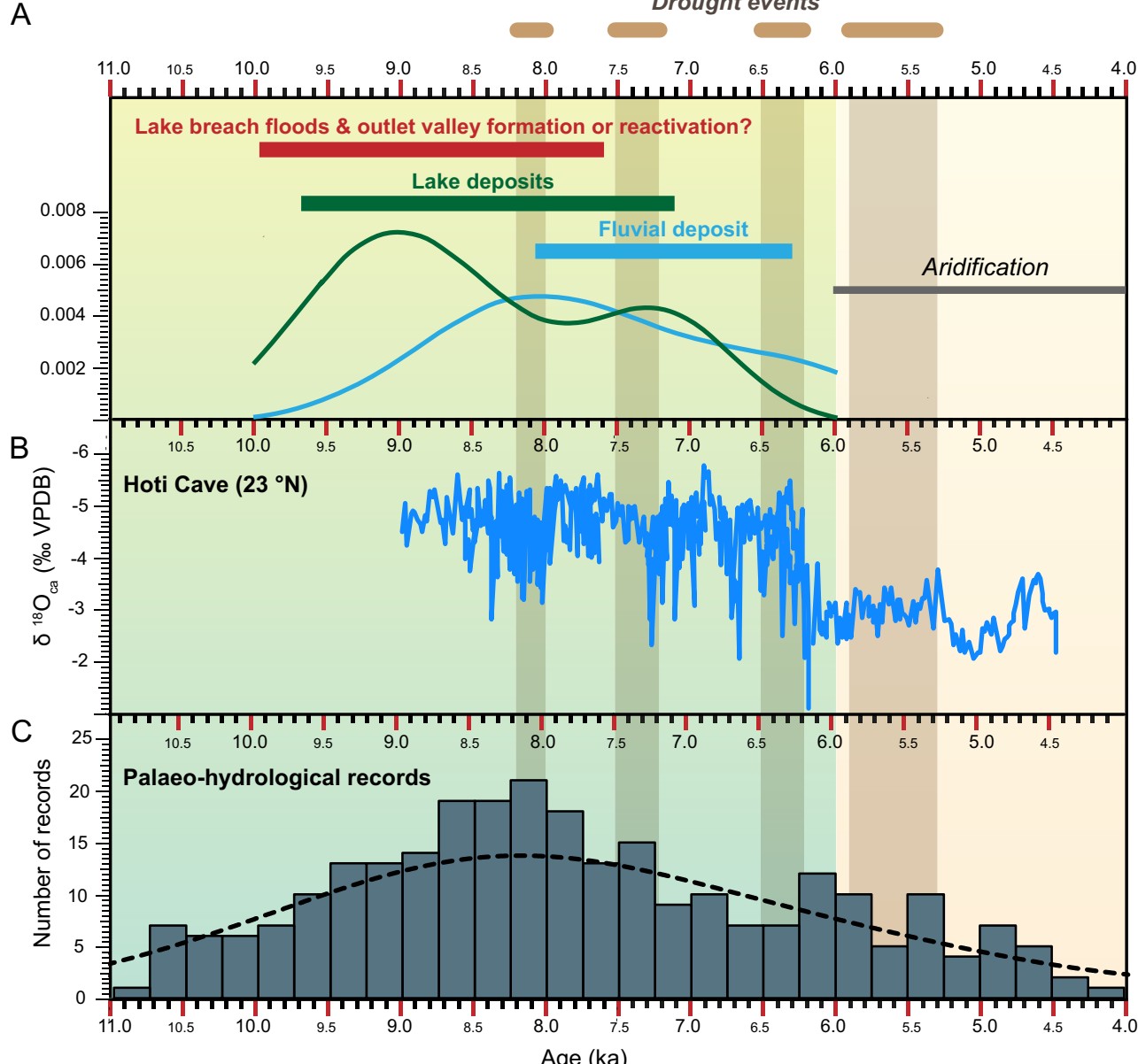

**Fig. 8 | Chronological data from lacustrine and fluvial deposits, alongside regional palaeoclimate records and hydrological indicators from the Arabian Peninsula. A** Chronological data showing the radiocarbon ages obtained from the lacustrine deposit (green), fluvial deposit (blue), and the possible age of the outlet valley (red) (Supplementary data 3). The curves were generated using kernel density estimation. **B** Hoti Cave $\delta^{18}O_{ca1}$ record, which indicates a significant humid decline at nearly 6.2 ka BP[3]. **C** A compilation of 397 OSL and radiocarbon ages was obtained from records indicative of hydrological changes, such as lacustrine, fluvial, springs, and palaeosols, from across the Arabian Peninsula (Supplementary data 1 and 2). This compilation shows that the HHP began at approximately 11 ka BP, and peaked from nearly 9.5 ka BP to nearly 7.2 ka BP.

erodibility, the slope of the terrain, or sediment load. Nonetheless, these factors mainly control the erosion rate, the shape of the valley, or the mode of sediment transport, while water overflow remains the primary mechanism in forming outlet valleys.

## Fingerprinting the African Monsoon on Arabian Desert landscapes

Placed in the debate of the African Monsoon's contribution to the greening of the Arabian Desert, our new reconstruction from floodplain lakes, a large lake segmented by a delta, and an approximately 150-km outlet valley offers a novel perspective on the extent and magnitude of this monsoon (Fig. 8). Our results indicate that the African Monsoon may have been strong or persistent enough to transport water and sediment over 1,000 km downstream, filling lakes and breaching one to carve out a large valley (Figs. 3, 4, 8,

and S10-S15). Such conditions likely occurred episodically, as inferred from the terraces within the outlet valley. This contradicts the conventional view of the greening of Arabia, where previous hydrologic and geomorphic studies suggest a weak African monsoon during the HHP or earlier Pleistocene humid periods[7]. Our observations from the lake breach and outlet valley resemble morphological features from various locations and time periods, such as the late Quaternary English Channel[28,39], the Mediterranean Sea after the Messinian Salinity Crisis[40], and the early Martian surface[29,41]. These analogies share similarities with our geomorphic observations, including an elongated valley, channels with streamlined islands, slope reversal, amphitheater-headed valleys, and terraces within the valley. This supports our interpretation that the 150-km-long outlet valley likely formed by the filling and breaching of the lake. This mechanism, which probably occurred during the HHP or earlier humid periods, indicates that the

African Monsoon played a key role in the repeated greening of Arabia. However, further details, such as the number of events involved, the timing of flows, and the mode of sediment transport, require in-situ observations, measurements, and precise absolute dating. The Indian Monsoon and the Mediterranean Westerlies may have contributed to the greening of Arabia[3,10,14,18,42,43]. However, the African Monsoon likely exerted a stronger influence, particularly on the Asir Mountains' drainage systems and the areas downstream. Due to their geographical distance, these regions were less affected by the Mediterranean Westerlies and the Indian Monsoon (Fig. 1). Together, these results show that the African Monsoon was capable of significant landscape transformations and likely created a habitable environment for human occupation during the Holocene, as shown by the presence of archaeological sites along the Wadi ad Dawasir floodplains in the Rub' al Khali Desert (Fig. 1)[11,12].

## Methods

### Site characterisation and sampling

We first explored the northern part of the Rub' al Khali region using an array of remote sensing data. This investigation incorporates observations and measurements from multi-spectral images from Landsat 8 (with a resolution of 30 metres per pixel), Sentinel-2 imagery (10 meters per pixel), and Google Earth imagery. The topographic data (i.e., elevation) was extracted from the Shuttle Radar Topography Mission (SRTM) with a resolution of 30 metres[44]. The reported vertical accuracy of the SRTM data points in open terrain is ±16 metres for 90% of the observations[45]. This ±16-metre margin represents the uncertainty in the absolute values of each pixel or cell. However, this level of uncertainty is unlikely to significantly affect the measurement of distances between objects. Examples include the 20-metre distance from the delta top to the breach point and the >20-metre thickness of inverted channels. A field investigation supplemented this remote analysis carried out in February 2022, where we employed unmanned aerial vehicle (UAV) imagery to study the regional topography, distinguish landforms, and collect in situ stratigraphic observations. Our field observations and samples are from the floodplain lakes that contributed to the main lake and its outlet valley. To sample and characterise the sedimentology of the palaeolake and palaeorivers, we excavated five sites featuring lacustrine depositional environments and three other sites marking fluvial environments (Figs. 1 and 2). We measured the sections and collected 24 samples at various intervals to constrain their age using radiocarbon and radiogenic isotopes to trace their provenance.

### Water volume calculations and lake-level modeling

We extracted approximately 73,166,782 data points from the lake and its surroundings using the SRTM digital elevation models. We assessed water volumes at two main geomorphic features conducive to the presence of a water body; 128 m asl, where we observed a fan-shaped landform, and 148 m asl, where we identified a breach point for the lake within the outlet valley. We employed the Delaunay Triangulation algorithm in Python to compute the volume of the enclosed area at the two elevations mentioned above. This algorithm is commonly used for these purposes and other similar calculations[46]. To delineate the lake's shorelines, we utilised the Water Level Tool in Global Mapper to compute the lake level at various elevations.

### Palaeohydraulic reconstructions

Climate models and geologic records can provide information on mean annual precipitation and temperature. However, to estimate the channel-forming discharge—which likely represents the maximum rainfall events required to either carve a new channel or fill an existing one to its brims—we used channel width (W) as a proxy for discharge. This was due to the fact that it was the only information regarding paleochannels we could not directly access in the field. We found a well-preserved drainage network that directly connects with the ancient lake. We measured the channel width within this drainage network at 29 sites (with a mean channel width of 230 ± 87 m; Figs. 6C and S21) and applied an empirical relationship (1)

developed by Eaton[47] to calculate the channel-forming discharge (m³/s) with a standard error of 29%.

$$Q_w = 0.10W^{1.866} \quad (1)$$

We subsequently calculated the drainage area (A) of these paleo-channels using Hack's Law (Eq. 2)[48]. Hack's Law posits a linear relationship between the drainage area (A) and the mainstream length (L), which, in our case, has a mean of 45 ± 9 km. Applying Eq. 2 yielded a mean drainage area of approximately 200 km², which is roughly similar to the manually delineated area of the system, averaging 228 ± 9 km².

$$L = 1.4A^{0.6} \quad (2)$$

To translate the discharge to rainfall intensity (m/s), we divided the discharge (Qw) by the drainage area (A), yielding a range of 17–78 mm/hr. This intensity is classified as heavy to violent[49]. However, our data cannot confirm whether such intensities could represent a single event or be spread over multiple events due to the absence of stratigraphic information for this drainage network.

### Automated terrain classification

To corroborate our observations derived from remote sensing of landforms within the hypothesized lacustrine depression, we utilized a pattern recognition-based algorithm known as "Geomorphons."[50] Unlike traditional methods that rely on differential geometry, Geomorphons classify landforms primarily through slope analysis, enabling the differentiation of features such as ridges, shoulders, slopes, valleys, spurs, and flat terrains. Our analysis focused on identifying flat terrains, indicative of terraces, by applying a slope threshold of 0.1°[51]. This threshold is designed to capture flat surfaces characteristic of both lacustrine terraces and global shelf terraces. The algorithm was executed on Shuttle Radar Topography Mission (SRTM) data within the System for Automated Geoscientific Analyses (SAGA) framework[52].

### Radiocarbon dating

We used radiocarbon dating on the extracted total organic carbon (TOC) from 17 fluvial, lacustrine, and mollusc samples (Supplementary data 3). The samples were collected from 8 cleaned fluvial and lacustrine sections. To eliminate the carbonate from the samples, we fumigated the samples in silver capsules with 37% HCl[53]. This was followed by neutralising the samples with NaOH for 48 hours at 65 °C to remove any residual acid. The samples were then measured as gas targets using the MICADAS system at ETH-Zurich, normalised using oxalic acid II (NIST SRM 4990 C), and corrections were made for any constant contamination introduced during fumigation and from the capsules. The radiocarbon ages in radiocarbon years were calibrated using the Calib 8.20 software[54]. The most recent calibration curve was used for the calibration, IntCal20[55], and results are reported in calendar years before present (cal BP) with a 1 sigma uncertainty range. Figures S8 and S9 show an example of a calibrated versus uncalibrated age, relative to the IntCal20 calibration curve, and the age distributions of all sample ages, respectively.

### Mollusk identification

We determined the species based on key external features such as shell shape, concentric growth lines, and overall shell thickness[24]. However, we acknowledge the absence of internal hinge views, which are crucial for definitive species identification. Despite this limitation, the external morphology provided sufficient grounds for the identification of both *Unio tigridis* and *Melanoides tuberculatus*.

### Calculating the potential energy released during the valley incision

To calculate the potential energy (PE) released during the incision of the outlet valley, we used an equation developed based on the assumption that

this energy is proportional to the volume of water ($V_F$) discharged during the breach flood[29]. This relationship can be mathematically formulated as follows:

$$PE = \rho g h V_F \qquad (3)$$

In the given equation, $\rho$ represents the density of water, conventionally assumed to be 1000 kg/m³. There is a high possibility for both large and fine materials to be mobilized during a lake breach. However, information about suspended sediments during such events is lacking, even for the most recent incidents. Therefore, we base our calculations on water density, which means our estimates of the potential energy released represent the minimum possible. The g denotes the acceleration due to gravity, approximated to be 9.81 m/s², and h refers to the breach depth, which, based on our measurements in this study, is determined to be 20 m. The breach depth ($h$) was determined by subtracting the water surface elevation before the breach[27], which is the water level of the lower delta zone (128 m asl), from the water surface elevation after the breach (148 m asl). The volume drained is calculated based on the difference in volumes between these two levels ($1.6 \times 10^{10}$ m³).

## Radiogenic isotopes

The radiogenic isotope composition of neodymium (Nd) and strontium (Sr) was measured on the siliciclastic fraction of the sediments. The five samples analysed (PA-21, PA-25, PA-27, PA-32, and PA-37B, Fig. 2) are composed of fine-sand to mud-sized marl-rich grains. They were processed as a whole in order to determine their bulk composition and potential source area. The samples were finely ground, washed with ultrapure (MilliQ) water and decarbonated using a buffered acetic acid solution before total dissolution. Subsamples of approximately 0.05 g of dried sediments were totally dissolved using concentrated hydrochloric, nitric, and hydrofluoric acids. Standard column-chromatography procedures were applied to separate and purify Nd and Sr[56–58]. The Nd and Sr isotope compositions were measured on a Thermo Neptune multi-collector inductively-coupled plasma mass spectrometer (MC-ICP-MS) at GFZ Potsdam. Blank levels were negligible for both radiogenic Nd and Sr isotopes ( < 0.4 ppb or <0.04% of sample size for Nd and <5 ppb or <0.5% of sample size for Sr). The isotope results were normalised to the accepted values of the JNdi standard for Nd ($^{143}$Nd/$^{144}$Nd = 0.512115)[59] and of the NIST SRM 987 standard for Sr ($^{86}$Sr/$^{87}$Sr = 0.710245). The Nd isotope ratios are reported as $\varepsilon_{Nd}$, which is the $^{143}$Nd/$^{144}$Nd ratio normalised to CHUR ($^{143}$Nd/$^{144}$Nd = 0.512638)[60] and multiplied by 10,000. External reproducibility was estimated by repeated measurements of SPEX and JNdi standards for Nd and an AA and the NIST SRM 987 standards for Sr. The standard 2σ uncertainty for each measurement session is reported and was ≤ 0.23 $\varepsilon_{Nd}$ units for Nd and < ±0.000023 for Sr.

## Transient Earth System model simulations

We use an ensemble of transient simulations for the last 21000 years performed in the Max Planck Institute Earth System Model version 1.2 (MPI-ESM)[61] and the Community Climate Model version 3 (CCSM3)[35]. The MPI-ESM simulation set includes the simulations „Ice6G_P2"[62], „Ice6G_P3"[63], „Glac1D_P2", and „Glac1D_P3" prepared by Kapsch et al.[64] that has been integrated with different ice-sheet and surface topography forcings (GLAC-1D or ICE-6G reconstructions[65–67] and slightly different model parameter tuning. It furthermore includes the simulation presented by Dallmeyer et al.[68,69] that used the GLAC-1D ice-sheet reconstruction as forcing. The model has been run from a quasi-equilibrium glacial climate state to modern climate (1950 CE) for all simulations. Greenhouse gas concentration[68] and insolation[69] have been prescribed according to reconstructions. Further details on the model setup and simulations are described in Kapsch et al.[64] and Dallmeyer et al.[68].

The TraCE-21ka simulation set contains the TraCE-21k[70] and TraCE-21k-II[71] simulation. Both have been performed in the CCSM3 model and span the last 22000 years. The model has been forced with variations in

greenhouse gas concentration[72], insolation[71], and continental ice sheets and coastlines according to the ICE5G reconstruction[73–76]. While in TraCE-21k-I, the meltwater flux to the Gulf of Mexico and the North Atlantic has been prescribed for the entire simulation, no freshwater flux, but a reconstructed AMOC strength has been prescribed for the Bølling-Allerød interstadial ( ~ 14.7–12.9 ka) and for the Holocene ( ~ 11.5 ka to the present) in TraCE-21k-II. For further details on the TraCE-21k simulations, see He[77], Liu et al.[72], and He and Clark[73].

All simulations ran at the spatial resolution T31 ( ~ 3.75°x3.75° on a Gaussian grid) but have been bilinearly interpolated on a regular 0.5° grid in this study.

## Palaeohydrological record compilation

In order to compare our dates with those from other sites that have recorded hydrological changes during the HHP in Arabia, we utilised 308 radiocarbon dates and 87 OSL dates from various hydrological environments. These environments include fluvial, lacustrine, springs, and paleosols situated in 113 different sites across the Arabian Peninsula. Most of the data was compiled by Woor[4], with additional ages collected from recent work by Neugebauer et al.[8]. We used these ages to statistically test the distinction between the development of fluvial and lacustrine environments across Arabia during the HHP using the Kolmogorov-Smirnoff test (Fig. S17; Supplementary data 1 and 2). The binning size is between 600 to 2000 years.

## Lithospheric flexural modeling

We used 2D thermo-mechanical geodynamic modeling as in (Muller et al., 2024; Fioraso et al., 2024; Stuwe et al., 2022; Sternai et al., 2021)[78–81] to estimate the variations of surface elevations induced by surface load changes related to lake filling and breaching. We use the present-day topography and impose lake water filling at 128, 148, and 202 m asl. The model domain is 500 km wide and 150 km deep and accounts for a uniform crust (with rheology of quartzite) as well as the mantle beneath (with rheology of dunite). The lake (load) is located in the center-top of the model and is 50 km wide. We compute the viscoelastic vertical deformation over 18,000 years (with 100 years timesteps), assuming linear changes in lake levels (constrained by our data). To assess the sensitivity of surface elevation changes to lake load changes we test six different lithospheric structures with crustal thicknesses of 20, 30, and 40 km and lithospheric thicknesses of 80 and 120 km. The max-min uplift rates over time are summarized in Fig. S19. We found that the present topography likely has been uplifted by 1.5 metres, possibly exposing the palaeoshorelines alongside water-formed landscapes in the area to more erosion.

## Limitations

To keep our findings conservative, we identify three main limitations that affect our palaeohydraulic calculations, the interpretations associated with the geomorphic and stratigraphic context of the main lake and its outlet valley, and the precise timing of the outlet valley incision. The first limitation stems from the lack of in situ measurements for the channels in the catchment area and the geometry of the outlet valley. Most measurements were derived from SRTM data and cross-validated using Google Earth engine and Landsat imagery. Fieldwork in the inferred lake and outlet valley was not conducted as it was not part of our initial field campaign due to logistics. However, we recognize the importance of direct sedimentological evidence and plan to conduct field investigations in this area in the future. The second limitation concerns the interpretations of the distant lake and its outlet valleys without direct in situ validation. Although we observe drainage networks connected to the depression and an outlet valley, suggesting that the depression was filled and subsequently breached to form a valley, our findings do not provide in situ evidence to support this hypothesis. Nonetheless, our study presents a plausible mechanistic model supported by landscape analysis and slope and elevation data from SRTM data. The third limitation is the lack of a chronological framework to determine the exact timing of the outlet valley formation. While our radiocarbon dating from the catchment area, along with previous OSL and radiocarbon dates from

nearby outcrops[5], suggests that the system was at least reactivated during the early to middle Holocene, further dating is necessary to establish when the main lake was filled and the outlet valley was formed.

Overall, while our findings provide insights into the extent and magnitude of the African Monsoons in the Arabian Desert, they are limited by the lack of field validation for the main lake and its outlet valley. Future work will focus on detailed field observations, further radiometric dating of the outlet valley incision, and in-situ characterization of the main lake and its outlet. These efforts will enhance our understanding of the region's geomorphology, paleoenvironment, and paleoclimate.

## Data availability

The SRTM digital elevation models, data points extracted for volume calculations, OSL and radiocarbon dates, and geochemical data have been deposited at https://doi.org/10.5281/zenodo.11685013

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

## Acknowledgements

The authors would like to thank Prof. Albert Matter for his constructive feedback on the first draft of this manuscript. We would like to express our gratitude to Thomas Kleinen, Uwe Mikolajewicz, and Marie Kapsch from the Max Planck Institute for Meteorology, and Feng He from the University of Wisconsin–Madison for providing MPI-ESM and TRACE-21k simulations, respectively. We also thank Dr. Morteza Djamali for his attempts to extract pollen and spores. This manuscript significantly benefited from the comments and suggestions of Dr Khalil Azennoud and four anonymous reviewers. Abdallah Zaki was funded by the Swiss National Science Foundation (project number: P500PN_206718). P.S. has been supported by Fondazione Cariplo and Fondazione CDP (Grant n° 2022- 1546_001), by the Italian MUR (Project Dipartimenti di Eccellenza, TECLA, Department of Earth and Environmental Sciences, University of Milano-Bicocca) and by the Alexander von Humbolt Foundation. We thank Elsevier for granting us a license to use the data to generate Fig. S13 (5759240915535).

## Author contributions

A.S.Z. and S.C. designed the research. A.S.Z., A.D., G.B., N.H., C.B., A.D., P.S., O.W., F.S., and H.K. performed the analysis and visualization. A.S.Z., A.D., G.B., M.S., M.P., F.B., A.M.A., and S.C. contributed to the interpretation of the results. All authors (A.S.Z., A.D., G.B., N.H., C.B., A.D., P.S., S.W., O.W., H.K., M.S., M.P., F.S., G.P., M.A., F.V.B., A.M.A., and S.C.) discussed the design, methods, and results, and contributed to the writing of the paper.

## Competing interests

The authors declare no competing interests.

## Additional information

¹Department of Earth Sciences, University of Geneva, Geneva, Switzerland. ²Division of Geological and Planetary Sciences, California Institute of Technology, Pasadena, CA, USA. ³Department of Earth and Planetary Sciences, Jackson School of Geosciences, The University of Texas at Austin, Austin, TX, USA. ⁴Physical Science and Engineering Division, King Abdullah University of Science and Technology, Thuwal, Saudi Arabia. ⁵Geological Institute, ETH Zürich, Zürich, Switzerland. ⁶GFZ German Research Centre for Geosciences, Climate Dynamics and Landscape Evolution, Potsdam, Germany. ⁷Max Planck Institute for Meteorology, Bundesstrasse 53, 20146 Hamburg, Germany. ⁸Department of Earth and Environmental Sciences, University of Milan-Bicocca, Milano, Italy. ⁹GFZ German Research Centre for Geosciences, Lithosphere Dynamics, Potsdam, Germany. ¹⁰Department of Geoscience, University of the Fraser Valley, Abbotsford, Canada. ¹¹Department of Earth, Ocean and Atmospheric Sciences, University of British Columbia, Vancouver, Canada. ¹²New York University, Tandon School of Engineering, Brooklyn 11201 NY, USA. ¹³Department of Geology, Alexandria University, Alexandria, Egypt. ¹⁴Université de Strasbourg, CNRS, Institut Terre et Environnement de Strasbourg, UMR 7063, 5 rue Descartes, Strasbourg F-67084, France. ¹⁵Australian Research Centre for Human Evolution, Griffith University, Brisbane, Australia. ¹⁶Human Origins Program, National Museum of Natural History, Smithsonian Institution, Washington, DC, USA. ¹⁷School of Social Science, University of Queensland, Brisbane, Australia. ¹⁸Aix-Marseille Université, CNRS, IRD, INRAE, CEREGE, Aix en Provence, Cerege, France. ¹⁹Department of Geography & GIS, Ain Shams University, Cairo 11566, Egypt. ✉e-mail: Abdallah.zaki@jsg.utexas.edu; abdalla.s.zaki@gmail.com

