## [Transparent Peer Review file · Communications Earth & Environment]

Monsoonal imprint on late Quaternary landscapes of the Rub' al Khali Desert

Corresponding Author: Dr Abdallah Zaki

This manuscript has been previously reviewed at another journal. This document only contains information relating to versions considered at Communications Earth & Environment.

Version 0:

Decision Letter:

Dear Dr Zaki,

Your revised manuscript titled "Monsoonal imprint on late Quaternary landscapes of the Rub' al Khali Desert" has now been seen by reviewer #3 and #4, and by a new reviewer (reviewer #5), whose comments appear below. In light of their advice we are delighted to say that we are happy, in principle, to publish a suitably revised version in Communications Earth & Environment.

We therefore invite you to revise your paper one last time to address the remaining concerns of our reviewers. At the same time we ask that you edit your manuscript to comply with our format requirements and to maximise the accessibility and therefore the impact of your work.

EDITORIAL REQUESTS:

****Please take care to match our formatting and policy requirements. We will check revised manuscript and return manuscripts that do not comply. Such requests will lead to delays. ****

SUBMISSION INFORMATION:

OPEN ACCESS:

Communications Earth & Environment is a fully open access journal. Articles are made freely accessible on publication. For further information about article processing charges, open access funding, and advice and support from Nature Research, please visit <https://www.nature.com/commsenv/open-access>

At acceptance, you will be provided with instructions for completing the open access licence agreement on behalf of all authors. This grants us the necessary permissions to publish your paper. Additionally, you will be asked to declare that all required third party permissions have been obtained, and to provide billing information in order to pay the article-processing

charge (APC).

Link Redacted

Best regards,

Carolina Ortiz Guerrero, Ph.D.
Associate Editor
Communications Earth & Environment

REVIEWERS' COMMENTS:

Reviewer #3 (Remarks to the Author):

The authors have spent much time and effort in addressing the numerous comments and concerns of all the reviewers. The manuscript has improved significantly as a result. The addition of a discussion on the limitations and uncertainties of the data and interpretations, inclusion of additional supporting figures, as well as a changed title all contributed to strengthening the revised manuscript.

Reviewer #4 (Remarks to the Author):

The revised manuscript is a significant improvement compared to the previous version submitted. The authors have considered the reviewers previous comments objectively and is better supported to present a much more balanced and coherent set of findings and explanations for these. The title is much better focussed for the research questions being addressed, the findings presented, and the interpretation of these. The paper now supports the major claims of the paper in a clear and logical manner. The previous version presented two fragmented sections on different components of the fluvial system and the likely evolution of these but without the necessary evidence to support this. The authors have also removed some contentious points raised, readdressed these and provided new supporting evidence to support the evidence presents especially regarding landform evolution in the system. The paper has now addressed these deficiencies and the rebuttal presented to the previous reviewers' comments have been considered carefully and corrected. The paper now presents a better understanding of the paleo development of the drainage catchment and evolution of the large depression and probable mechanisms leading to this and the feeder channels and channel overflow. I am pleased to see that the authors' now state that whilst the chronological evidence is Holocene in age that the system is likely to be much older in age in terms of its origins, and that the likelihood of multiple, episodic flooding events (rather than one event) is most likely for the evolution of the overflow channel. Whilst not all of the points raised can be addressed the authors' acknowledge some of the limitations presented and how these can be addressed in the future in a reassuring and convincing manner.

The work is reproducible and the re-analyses conducted are robust (within the limitations of the field evidence and methods employed). I believe that the manuscript is now acceptable for publication subject to the minor changes proposed below help to improve some of the terms used for clarity.

Abstract

Page 2, Line 48 - suggest replacing '~1100 km² lake' with '~1100 km² topographic depression'

Page 2, Line 51 suggest replacing 'dotted' with 'punctuated'

Page 2, Lines 51-53 – suggest replacing 'Analysis of lacustrine and fluvial sedimentary deposits enabled us to track sediment routing over distances extending 1000 km from the Asir Mountains.' with 'Analysis of lacustrine and fluvial sedimentary deposits implies sediment routing across distances of up to 1000 km from the Asir Mountains.'

Main

Page 4, Lines 108-109. Rephrase sentence 'The Rub' al Khali Desert is a large sedimentary basin, located downstream of the Asir Mountains (Fig. 1), characterized by aeolian dunes, drainage systems, and lakes.' This infers that lakes exist today – which they do not. Suggest the following revisions: 'The Rub' al Khali Desert is a large sedimentary basin, located downstream of the Asir Mountains (Fig. 1), characterized by aeolian dunes, drainage systems, and paleo lacustrine and palustrine deposits.'

Page 4, Lines 114-115. Replace 'lake and river deposits' with 'lacustrine, palustrine and fluvial deposits'. Note there are no perennial river deposits or extant lakes today. Enzel et al., 2015 make a case for the differences between lakes and wetland deposits from a terminological perspective based on geomorphic characteristics. Whilst there may be evidence for some former paleo lakes, not all deposits described in the paper or shown in the imagery will necessarily fall within these parameters, hence cautious use of palustrine and lacustrine is more acceptable. In addition, as the drainage systems are

ephemeral in nature it would be better to describe them as fluvial wadi systems rather than river deposits per se. Enzel, Y, Kushnir, Y, and Quade, J. "The middle Holocene climatic records from Arabia: Reassessing lacustrine environments, shift of ITCZ in Arabian Sea, and impacts of the southwest Indian and African monsoons." *Global and Planetary Change* 129 (2015): 69-91.

Page 4, Line 120. Consider replacing 'filling and breaching of a downstream lake....' with 'filling and breaching of a large downstream topographic depression by water....'

Geologic evidence of water-formed landscapes in the Rub'al Khali Desert

Page 5, line 130. Consider replacing 'rivers and lakes' with 'fluvial, lacustrine and palustrine systems'.

Page 5, line 141 and 142. Replace 'wetland' with 'palustrine' as this term covers wetlands, marshes, swamps etc..

Page 6, line 178 change to '....suggest a palaeolandscape composed of floodplains interspersed with waterbodies including lakes, ponds and wetlands a few tens of square kilometers in area.'

Page 6, line 187 insert 'wetlands' so as to read ' lakes, wetlands, and fluvial systems...'

Geomorphologic evidence of lake development and breach-induced flooding

Page 8, line 208. Inset 'We interpret the depression to host large water body forming a lake....'

Page 8, line 226-228. change to 'However, there is a possibility that these cliffs were originally formed as a result of regional tectonic uplift, and subsequently modified by water and wind activities.'

Time-space evolution of lake systems

Page 13, line 302. Should this read ' 128, 141 and 148 m asl' rather than '141, 128 and 147 m asl' ?

Page 13, line 327. '....this could be related to carbon contamination'. The age inversions could be related to reworking of older sediment into younger sediments giving rise to the discrepancies noted.

Page 13, line 333 should 'ended' read 'ending'?

Reviewer #5 (Remarks to the Author):

Dear editor and authors

After carefully reviewing the revised manuscript by Zaki et al., I can confirm that the revised version addresses most of the reviewers' comments, resulting in significant improvements.

The manuscript investigates the impact of the African Monsoon during the Holocene Humid Period on the landscapes of the Rub' al Khali Desert in Saudi Arabia. Using a multidisciplinary approach, including remote sensing, field observation, some radiogenic isotopic tracers, and climate model outputs, the study reconstructs a paleolandscape shaped by intense precession-paced monsoonal rainfall (dominance of the African Monsoon). While the findings offer valuable insights into the role of the African Monsoon in driving environmental changes and influencing potential human occupation in the region, some areas require further clarification and discussion, which I outline below.

In lines 96–99, the authors appropriately acknowledge the contributions of the Indian Monsoon and Mediterranean Westerlies alongside the African Monsoon during the HHP (I believe it is in response to one of Reviewer #4's comment); evidence of enhanced Mediterranean-sourced winter precipitation during low precession periods (enhanced W African Monsoon) in North Africa was reported from speleothem (Ait Brahim et al., 2023; Rogerson et al., 2019) and lake sediment (Cheddadi et al., 2021) records. However, throughout the manuscript, the African Monsoon is consistently favored, albeit with an implicit preference as the dominant influence. While this is supported to some extent, the manuscript does not explicitly address how the contributions of the Indian Monsoon or Mediterranean Westerlies were (quantitatively) ruled out, leaving some ambiguity about the relative magnitudes of these synoptic systems' influences. Including additional (stable) isotopic data directly contrasting these influences would strengthen the interpretation. Alternatively, a more focused discussion on the interplay or separation of these atmospheric systems would enhance the clarity and robustness of the conclusions.

Ait Brahim, Y., Sha, L., Wassenburg, J.A., Azennoud, K., Cheng, H., Cruz, F.W., Bouchaou, L., 2023. The spatiotemporal extent of the Green Sahara during the last glacial period. *iScience* 26. <https://doi.org/10.1016/j.isci.2023.107018>

Cheddadi, R., Carré, M., Nourelbait, M., François, L., Rhoujjati, A., Manay, R., Ochoa, D., Schefuß, E., 2021. Early Holocene greening of the Sahara requires Mediterranean winter rainfall. *Proceedings of the National Academy of Sciences* 118, e2024898118. <https://doi.org/10.1073/pnas.2024898118>

Rogerson, M., Dublyansky, Y., Hoffmann, D.L., Luetscher, M., Töchterle, P., Spötl, C., 2019. Enhanced Mediterranean water cycle explains increased humidity during MIS 3 in North Africa. *Climate of the Past* 15, 1757–1769. <https://doi.org/10.5194/cp-15-1757-2019>

I am curious why the authors did not conduct fieldwork in the area of the inferred large lake and its outlet valley, focusing instead on the upstream catchment area. Direct sedimentological evidence, such as an analysis of the inferred delta deposits and terraces, would have significantly strengthened the interpretation of the lake's occurrence and its subsequent breaching. In addition, with robust chronological controls, it is likely that evidence of previous precession-paced intense monsoon influences (e.g., MIS 5a, 5e, and others) can be identified there.

While the authors acknowledge the lack of field observations in the section on limitations (following previous reviewers'

comments), I believe they should objectively provide an explanation for this omission, whether due to logistical challenges, access difficulties, or other reasons.

Figs. 2, S4. I have some doubts regarding the stratigraphic section in Fig. 2, as it appears to correspond the outcrop in Fig. S4A and B if we look at the given coordinates. However, the lithofacies shown in both the stratigraphic section and the field view do not match.

Could you please add more detail on the lithological significance of the laminated deposits in the caption of Fig. S4A and B? Are you interpreting these as fluvial or lacustrine deposits?

L76. For more accurate terminology, the Early Holocene (Greenlandian) spans the period between 11.7 and 8.2 cal. ka BP. Please correct.

L80. Late Quaternary is not chronostratigraphically defined. You may refer to Late Pleistocene to Holocene, or explicitly mention a time period (e.g., past 130 ka or so).

L81. If you are referring to chronostratigraphically defined periods, please capitalize Early and Middle (Early Holocene, Middle Holocene).

L148. Please specifically refer to which panel in Fig. S4 (here I believe it's Fig. S4C).

L150. Again, please specify which panel. Here, I believe you should refer to Fig. S4A and B.

L150-151. If by "wetlands" you mean marshes, swamps, fens, or bogs, I don't believe these environments typically produce carbonate-rich deposits. Rather, they are generally characterized by organic sedimentation with frequent vegetation remains.

L161-166. I would ask the authors how they ruled out the possibility that both facies (associations) belong to the same fluvial regime. We often find both fine- and coarse-grained clastic sediments deposited simultaneously in different fluvial sub-environments, e.g., overbank (floodplain, crevasse splays, natural levees), channel bars, oxbow 'lakes'. Lateral facies shifts are quite common in braided fluvial settings.

L242. Please define "m asl" the first time it appears in the manuscript.

L394. Be consistent in using age unit: ka or kyr; a or yr.

I hope these revisions are helpful.

Sincerely

Khalil Azennoud

Author Rebuttals to Initial Comments:

Referee comments are in **bold black text**

Our replies to comments are in **purple text (Times New Roman)**

Text that we revised upon reflecting on one or more Referee comments is in **light blue text (Arial)**

Reviewer #3 (Remarks to the Author):

The authors have spent much time and effort in addressing the numerous comments and concerns of all the reviewers. The manuscript has improved significantly as a result. The addition of a discussion on the limitations and uncertainties of the data and interpretations, inclusion of additional supporting figures, as well as a changed title all contributed to strengthening the revised manuscript.

We thank Reviewer #3 for their positive feedback and previous comments, which substantially improved the manuscript.

Reviewer #4 (Remarks to the Author):

The revised manuscript is a significant improvement compared to the previous version submitted. The authors have considered the reviewers previous comments objectively and is better supported to present a much more balanced and coherent set of findings and explanations for these. The title is much better focussed for the research questions being addressed, the findings presented, and the interpretation of these. The paper now supports the major claims of the paper in a clear and logical manner. The previous version presented two fragmented sections on different components of the fluvial system and the likely evolution of these but without the necessary evidence to support this. The authors have also removed some contentious points raised, readdressed these and provided new supporting evidence to support the evidence presents especially regarding landform evolution in the system. The paper has now addressed these deficiencies and the rebuttal presented to the previous reviewers' comments have been considered carefully and corrected. The paper now presents a better understanding of the paleo development of the drainage catchment and evolution of the large depression and probable mechanisms leading to this and the feeder channels and channel overspill. I am pleased to see that the authors' now state that whilst the chronological evidence is Holocene in age that the system is likely to be much older in age in terms of its origins, and that the likelihood of multiple, episodic flooding events (rather than one event) is most likely for the evolution of the overspill channel. Whilst not all of the points raised can be addressed the authors' acknowledge some of the limitations presented and how these can be addressed in the future in a reassuring and convincing manner.

The work is reproducible and the re-analyses conducted are robust (within the limitations of the field evidence and methods employed). I believe that the manuscript is now acceptable for publication subject to the minor changes proposed below help to improve some of the terms used for clarity.

We thank Reviewer #4 for their positive feedback and for recognizing the robustness and reproducibility of our work. We have addressed the minor changes you suggested as follows:

Page 2, Line 48 - suggest replacing ‘~1100 km² lake’ with ‘~1100 km² topographic depression’

Thank you. Replaced.

“During the peak of the Holocene Humid Period (HHP) or before, intense rainfall reactivated alluvial floodplains and filled a ~1100 km² topographic depression, which eventually breached, carving a deep ~150 km-long valley.”

Page 2, Line 51 suggest replacing ‘dotted’ with ‘punctuated’

Fixed. Thank you

“Coupling geologic reconstructions with transient Earth system model simulations shows that this hydrological activity was linked to higher seasonal precipitation punctuated by repeated heavy events.”

Page 2, Lines 51-53 – suggest replacing ‘Analysis of lacustrine and fluvial sedimentary deposits enabled us to track sediment routing over distances extending 1000 km from the Asir Mountains.’ with ‘Analysis of lacustrine and fluvial sedimentary deposits implies sediment routing across distances of up to 1000 km from the Asir Mountains.’

Replaced. Thank you

“Analysis of lacustrine and fluvial sedimentary deposits implies sediment routing across distances of up to 1000 km from the Asir Mountains.”

Page 4, Lines 108-109. Rephrase sentence ‘The Rub’ al Khali Desert is a large sedimentary basin, located downstream of the Asir Mountains (Fig. 1), characterized by aeolian dunes, drainage systems, and lakes.’ This infers that lakes exist today – which they do not. Suggest the following revisions: ‘The Rub’ al Khali Desert is a large sedimentary basin, located downstream of the Asir Mountains (Fig. 1), characterized by aeolian dunes, drainage systems, and paleo lacustrine and palustrine deposits.’

We thank the reviewer for this suggestion. We replaced it.

“The Rub’ al Khali Desert is a large sedimentary basin, located downstream of the Asir Mountains (Fig. 1), characterized by aeolian dunes, drainage systems, and paleo lacustrine and palustrine deposits^{5,12}. The Asir Mountains are hypothesized to have been primarily fed by the African Monsoons⁵. Consequently, the drainage systems that originated and eroded upstream and deposited sediments downstream could serve as a proxy to constrain the magnitude and extent of the African Monsoon's influence on the Arabian Desert landscapes.”

Page 4, Lines 114-115. Replace ‘lake and river deposits’ with ‘lacustrine, palustrine and fluvial deposits’. Note there are no perennial river deposits or extant lakes today. Enzel et al., 2015 make a case for the differences between lakes and wetland deposits from a terminological perspective based on geomorphic characteristics. Whilst there may be evidence for some former paleo lakes, not all deposits described in the paper or shown in the imagery will necessarily fall within these parameters, hence cautious use of palustrine and lacustrine is more acceptable. In addition, as the drainage systems are ephemeral in nature it would be better to describe them as fluvial wadi systems rather than river deposits per se.

We thank the reviewer for this suggestion. We fixed it.

“Here we reconstruct an ancient water-sculpted landscape consisting of lacustrine, palustrine and fluvial deposits, coupled with a large outlet valley in the Rub’ al Khali Desert of Saudi Arabia (Fig. 1).”

Page 4, Line 120. Consider replacing ‘filling and breaching of a downstream lake....’ with ‘filling and breaching of a large downstream topographic depression by water....’

Replaced. Thank you.

“Our results illustrate the functioning of a large-scale source-to-sink fluvial system impacted by intense monsoons during the HHP, filling and breaching of a downstream topographic depression by water, and incision of a large outlet canyon.”

Geologic evidence of water-formed landscapes in the Rub'al Khali Desert

Page 5, line 130. Consider replacing ‘rivers and lakes’ with ‘fluvial, lacustrine and palustrine systems’.

Replaced. Thank you.

“Due to its location downstream of large ancient drainage systems such as Wadi ad-Dawasir and Wadi Sahba (Fig. 1; Figs. S1, S2), the Rub' al Khali Desert is a sensitive environment that records past hydroclimatic shifts in Arabia across various depositional environments, including fluvial, lacustrine and palustrine systems.”

Page 5, line 141 and 142. Replace ‘wetland’ with ‘palustrine’ as this terms covers wetlands, marshes, swamps etc..

Replaced. Thank you.

“Therefore, we interpret the region as preserving three different sedimentary depositional environments: aeolian, based on the complex linear dunes; fluvial, based on the incised and inverted channel fills and belts; and either lacustrine or palustrine, based on the eroded whitish mesas. Here, we focus on both the fluvial and lacustrine or wetland environments.”

Page 6, line 178 change to ‘....suggest a palaeolandscape composed of floodplains interspersed with waterbodies including lakes, ponds and wetlands a few tens of square kilometers in area.’

Changed. Thank you.

“Collectively, observations from large-scale satellite and aerial imagery, coupled with outcrop-scale observations, suggest a palaeolandscape composed of floodplains interspersed with waterbodies including lakes, ponds and wetlands a few tens of square kilometers in area.”

Page 6, line 187 insert ‘wetlands’ so as to read ‘ lakes, wetlands, and fluvial systems...’

Added. Thank you.

“The co-occurrence of these species points to a dynamic palaeolandscape where floodplain lakes, wetlands, and fluvial systems fluctuated with shifting precipitation patterns, providing critical habitats during humid periods. “

Geomorphologic evidence of lake development and breach-induced flooding

Page 8, line 208. Inset ‘We interpret the depression to host large water body forming a lake....

Thank you. Inserted.

“We interpret the depression to host large water body forming a lake as it is bordered by a distributary fan and connected to a valley ~150 km in length (Fig. 3).”

Page 8, line 226-228. change to ‘However, there is a possibility that these cliffs were originally formed as a result of regional tectonic uplift, and subsequently modified by water and wind activities.’

Thank you. Changed.

Time-space evolution of lake systems

Page 13, line 302. Should this read ‘ 128, 141 and 148 m asl’ rather than ‘141, 128 and 147 m asl’ ?

Yes. We fixed it.

Page 13, line 327. ‘....this could be related to carbon contamination’. The age inversions could be related to reworking of older sediment into younger sediments giving rise to the discrepancies noted.

Modified. Thank you.

“Although two lacustrine sections (sections 4 and 5 in Fig. 2) show older ages superimposed on younger ones, this discrepancy could be due to carbon contamination, which may have led to the reworking of older sediments into younger deposits.”

Page 13, line 333 should ‘ended’ read ‘ending’?

Fixed. Thank you.

Reviewer #5 (Remarks to the Author):

Dear editor and authors

After carefully reviewing the revised manuscript by Zaki et al., I can confirm that the revised version addresses most of the reviewers' comments, resulting in significant improvements.

The manuscript investigates the impact of the African Monsoon during the Holocene Humid Period on the landscapes of the Rub' al Khali Desert in Saudi Arabia. Using a multidisciplinary approach, including remote sensing, field observation, some radiogenic isotopic tracers, and climate model outputs, the study reconstructs a paleolandscape shaped by intense precession-paced monsoonal rainfall (dominance of the African Monsoon). While the findings offer valuable insights into the role of the African Monsoon in driving environmental changes and influencing potential human occupation in the region, some areas require further clarification and discussion, which I outline below.

Thank you, Dr. Azennoud, for your constructive feedback and for acknowledging the improvements in our manuscript. We have addressed the areas requiring further clarification in the revised version, as outlined in our response below

In lines 96–99, the authors appropriately acknowledge the contributions of the Indian Monsoon and Mediterranean Westerlies alongside the African Monsoon during the HHP (I believe it is in response to one of Reviewer #4's comment); evidence of enhanced Mediterranean-sourced winter precipitation during low precession periods (enhanced W African Monsoon) in North Africa was reported from speleothem (Ait Brahim et al., 2023; Rogerson et al., 2019) and lake sediment (Cheddadi et al., 2021) records. However, throughout the manuscript, the African Monsoon is consistently favored, albeit with an implicit preference as the dominant influence. While this is supported to some extent, the manuscript does not explicitly address how the contributions of the Indian Monsoon or Mediterranean Westerlies were (quantitatively) ruled out, leaving some ambiguity about the relative magnitudes of these synoptic systems' influences. Including additional (stable) isotopic data directly contrasting these influences would strengthen the interpretation. Alternatively, a more focused discussion on the interplay or separation of these atmospheric systems would enhance the clarity and robustness of the conclusions.

Thank you for this insightful comment. While both the Mediterranean Westerlies and the Indian Monsoon may have contributed to the greening of the Arabian Desert, the African Monsoon likely had the strongest influence on the formation and reactivation of Wadi ad-Dawasir. This is because the wadi is located beyond the primary reach of the other two monsoon systems. We have clarified this in the revised discussion.

“Fingerprinting the African Monsoon on Arabian Desert landscapes

*Placed in the debate of the African Monsoon's contribution to the greening of the Arabian Desert, our new reconstruction from floodplain lakes, a large lake segmented by a delta, and an approximately 150-km outlet valley offers a novel perspective on the extent and magnitude of this monsoon. Our results indicate that the African Monsoon may have been strong or persistent enough to transport water and sediment over 1,000 km downstream, filling lakes and breaching one to carve out a large valley (Figs. 3, 4, S10-S15). Such conditions likely occurred episodically, as inferred from the terraces within the outlet valley. This contradicts the conventional view of the greening of Arabia, where previous hydrologic and geomorphic studies suggest a weak African monsoon during the HHP or earlier Pleistocene humid periods⁷. Our observations from the lake breach and outlet valley resemble morphological features from various locations and time periods, such as the late Quaternary English Channel^{39,40}, the Mediterranean Sea after the Messinian Salinity Crisis⁴¹, and the early Martian surface^{29,42}. These analogies share similarities with our geomorphic observations, including an elongated valley, channels with streamlined islands, slope reversal, amphitheater-headed valleys, and terraces within the valley. This supports our conclusion that the 150-km-long outlet valley likely formed by the filling and breaching of the lake. This mechanism, which probably occurred during the HHP or earlier humid periods, indicates that the African Monsoon played a key role in the repeated greening of Arabia. However, further details, such as the number of events involved, the timing of flows, and the mode of sediment transport, require in-situ observations, measurements, and precise absolute dating. **The Indian Monsoon and the Mediterranean Westerlies may have contributed to the greening of Arabia^{3,10,17,43,44}. However, the African Monsoon likely exerted a stronger influence, particularly on the Asir Mountains' drainage systems and the areas downstream. Due to their geographical distance, these regions were less affected by the Mediterranean Westerlies and the Indian Monsoon (Fig. 1). Together, these results show that the African Monsoon was capable of significant landscape transformations and likely created a habitable environment for human occupation during the Holocene, as shown by the presence of archaeological sites along the Wadi ad Dawasir floodplains in the Rub' al Khali Desert (Fig. 1)^{11,12}.***

I am curious why the authors did not conduct fieldwork in the area of the inferred large lake and its outlet valley, focusing instead on the upstream catchment area. Direct sedimentological evidence, such as an analysis of the inferred delta deposits and terraces, would have significantly strengthened the interpretation of the lake's occurrence and its subsequent breaching. In addition, with robust chronological controls, it is likely that evidence of previous precession-paced intense monsoon influences (e.g., MIS 5a, 5e, and others) can be identified there.

While the authors acknowledge the lack of field observations in the section on limitations (following previous reviewers' comments), I believe they should objectively provide an explanation for this omission, whether due to logistical challenges, access difficulties, or other reasons.

Thank you for raising this comment. In response, we have addressed this in the following paragraph within the Limitations section.

“Limitations

*To keep our findings conservative, we identify three main limitations that affect our palaeohydraulic calculations, the interpretations associated with the geomorphic and stratigraphic context of the main lake and its outlet valley, and the precise timing of the outlet valley incision. The first limitation stems from the lack of in situ measurements for the channels in the catchment area and the geometry of the outlet valley. Most measurements were derived from SRTM data and cross-validated using Google Earth engine and Landsat imagery. **Fieldwork in the inferred lake and outlet valley was not conducted as it was not part of our initial field campaign due to logistics. However, we recognize the importance of direct sedimentological evidence and plan to conduct field investigations in this area in the future.** The second limitation concerns the interpretations of the distant lake and its outlet valleys without direct in situ validation. Although we observe drainage networks connected to the depression and an outlet valley, suggesting that the depression was filled and subsequently breached to form a valley, our findings do not provide in situ evidence to support this hypothesis. Nonetheless, our study presents a plausible mechanistic model supported by landscape analysis and slope and elevation data from SRTM data. The third limitation is the lack of a chronological framework to determine the exact timing of the outlet valley formation. While our radiocarbon dating from the catchment area, along with previous OSL and radiocarbon dates from nearby outcrops⁵, suggests that the system was at least reactivated during the early to middle Holocene, further dating is necessary to establish when the main lake was filled and the outlet valley was formed.*

Overall, while our findings provide insights into the extent and magnitude of the African Monsoons in the Arabian Desert, they are limited by the lack of field validation for the main lake and its outlet valley. Future work will focus on detailed field observations, further radiometric dating of the outlet valley incision, and in-situ characterization of the main lake and its outlet. These efforts will enhance our understanding of the region’s geomorphology, paleoenvironment, and paleoclimate.”

Figs. 2, S4. I have some doubts regarding the stratigraphic section in Fig. 2, as it appears to correspond the outcrop in Fig. S4A and B if we look at the given coordinates. However, the lithofacies shown in both the stratigraphic section and the field view do not match. Could you please add more detail on the lithological significance of the laminated deposits in the caption of Fig. S4A and B? Are you interpreting these as fluvial or lacustrine deposits?

Thank you for your comment. The field photographs in Fig. S4A and B do not correspond to the stratigraphic section shown in Fig. 2, as they represent different locations. We only included sites where samples were collected for dating and radiogenic isotope analysis, and their coordinates do not match those in Fig. 2. That said, we have added an interpretation in the caption of Fig. S4A and B, clarifying the lithological significance of the laminated deposits and whether they are of fluvial or lacustrine origin.

L76. For more accurate terminology, the Early Holocene (Greenlandian) spans the period between 11.7 and 8.2 cal. ka BP. Please correct.

We disagree with this comment, as some of the rivers and lakes studied in our paper remained active beyond 8.2 ka BP. Aridity began around 11 ka, so we believe that referring to the

early to middle Holocene accurately captures the event we focused on in this paper. Thank you. Please see our detailed response below.

“During the early Holocene to Middle Holocene (~11 to ~5.5 ka BP), perennial lakes and extensive drainage networks characterised Arabian landscapes^{5,6,7,8}.”

L80. Late Quaternary is not chronostratigraphically defined. You may refer to Late Pleistocene to Holocene, or explicitly mention a time period (e.g., past 130 ka or so).

Fixed. Thank you.

Increased rainfall during the early Holocene, driven by the northward expansion of African and Indian Ocean monsoon rain belts^{9,10,11}, is the most recent of the multiple wetter periods that have occurred synchronously with insolation changes associated with orbital precessional cycles throughout Late Pleistocene to Holocene.

L81. If you are referring to chronostratigraphically defined periods, please capitalize Early and Middle (Early Holocene, Middle Holocene).

We did not make this change, as we use 'early' and 'middle Holocene' in a more descriptive sense rather than as formal chronostratigraphic divisions, given that our ages deviate slightly from the known chronostratigraphically defined periods. Thank you.

L148. Please specifically refer to which panel in Fig. S4 (here I believe it's Fig. S4C).

Fixed. Thank you.

L150. Again, please specify which panel. Here, I believe you should refer to Fig. S4A and B.

Fixed. Thank you.

L150-151. If by “wetlands” you mean marshes, swamps, fens, or bogs, I don't believe these environments typically produce carbonate-rich deposits. Rather, they are generally characterized by organic sedimentation with frequent vegetation remains.

Thank you for raising this comment. In Figure S5 (please see below), we show trace fossils found atop the lacustrine deposits. These features bear a resemblance to *Cruziana*, *Taenidium*, and *Scoyenia*, suggesting that the site was either directly on the shoreline or in its immediate vicinity, reflecting the interaction between organisms and sediment in a transitional water-land environment, i.e., marshes, swamps.

Additionally, floodplain lakes and wetlands that support marshes and swamps have been documented by Matter et al. in nearby sites, which aligns with our interpretation. For further details, please refer to the reference provided below.

Matter, A. *et al.* Reactivation of the Pleistocene trans-Arabian Wadi ad Dawasir Fluvial System (Saudi Arabia) during the Holocene humid phase. *Geomorphology* **270**, 88–101 (2016).

L161-166. I would ask the authors how they ruled out the possibility that both facies (associations) belong to the same fluvial regime. We often find both fine- and coarse-grained clastic sediments deposited simultaneously in different fluvial sub-environments, e.g., overbank (floodplain, crevasse splays, natural levees), channel bars, oxbow ‘lakes’. Lateral facies shifts are quite common in braided fluvial settings.

Thank you for raising this comment. Our investigation relied on vertical facies changes, which indicate a systematic transition rather than a lateral shift between distinct sub-environments. The sedimentological characteristics (e.g., grain size trends) support a coherent depositional evolution within a single fluvial system. For further clarification, please refer to Example 6 in Fig. 2. L242. Please define “m asl” the first time it appears in the manuscript.

L242. Please define “m asl” the first time it appears in the manuscript.

It’s already defined.

“We calculated the volume of water required to fill the lake to the level of the lower zone delta (128 m above sea level (asl)), delta topset (141 m asl), and the lower incision level at the breach (148 m asl; Figs. S12 and S15).”

L394. Be consistent in using age unit: ka or kyr; a or yr.

The choice of age units is based on the dating technique used. For instance, we report ages in cal yr BP for radiocarbon dates, whereas OSL ages are typically reported in ka. This distinction aligns with standard conventions in geochronology.

I hope these revisions are helpful.

Thank you. They were helpful comments.